
# On the peculiar polarimetric signatures backscattered by a still or quasi-still wind turbine acquired by an X-band radar in stare mode at high temporal resolution (64 ms): preliminary investigations

Marco Gabella[1], Martin Lainer[1], Daniel Wolfensberger[1], and Jacopo Grazioli[2]

[1]Federal Office of Meteorology and Climatology MeteoSwiss, Locarno-Monti, CH-6605, Switzerland
[2]Environmental Remote Sensing Laboratory, École polytechnique fédérale de Lausanne, Lausanne, Switzerland

*Correspondence to*: Marco Gabella (marco.gabella@meteoswiss.ch)

**Abstract.** A still wind turbine (WT) observed with a fixed pointing radar antenna shows peculiar polarimetric signatures: during two minutes (from 17:08 to 17:10 UTC) of the first day (March 4, 2020) of the WT MeteoSwiss X-band radar campaign in Schaffhausen, the copolar correlation coefficient between the two orthogonal polarization states was persistently equal to 1. The reflectivity at vertical polarization was bounded between 38.5 and 41.5 dBz; however, the changes between two consecutive 64 ms values (retrieved by means of 128 transmitted pulses) were either 0 dBz or ±0.5 dBz. The 2-min median (mean) value was 40.0 (39.9) dBz over the 1875 echoes of this interval. The reflectivity at horizontal polarization was persistently equal to 56.5 dBz, which means no change exceeding ±0.25 dBz. The standard deviation (1874 degrees of freedom) of the differential phase shift, which in the absence of precipitation was, in fact, coincident with the dispersion of the differential backscattering phase shift, was as small as 3.0°. During two 10-min intervals (17:10-17:20 UTC and 17:30-17:40 UTC) the rotor has moved less than 1 revolution; however, this slow movement together with a change in blade angles and nacelle orientation was sufficient to cause large changes and significant variability in the polarimetric signatures, with two pairs of ZH consecutive values reaching the extreme of 78.5 dBz, which is the absolute reflectivity maximum reached in the whole campaign (March 4-21, 2020). Between 17:20 and 17:30 UTC, the rotor has accomplished 22.5 revolutions: the variability becomes smoother and softer in the central part of the interval (probably thanks to uniform rotor speed and "frozen" blade angles and nacelle orientation). It is desirable and recommended to extend this preliminary (32-minute) analysis (based on thirty thousand polarimetric measurables) to several other 10-min intervals with zero rotor speed.

## 1 Introduction

Wind turbines can heavily affect several types of sensitive and relevant radar observations including weather, surveillance, precision approaching and air traffic control radars. Further, the operation of air traffic radio navigation systems like VOR (VHF omnidirectional range) can be disturbed by nearby wind turbines (e.g., Morlaas et al., 2008; Douvenot et al., 2017). In 2021 the European countries invested about €41billion in new wind farms, covering 24.6 GW of new capacity (Brindley, 2022). However this is still far off from the European goal to reach its new climate change and energy security targets.



Consequently, the continuous and strengthened expansion of wind farms is of major concern for the weather (e.g., Norin, 2017) and aviation radar community (e.g., Cuadra et al., 2019). Wind turbines are large objects with a variety of movement patterns, which makes them a heavy source of clutter that is difficult to filter. Several studies exist in the literature regarding the impact of wind turbines on radar systems. From a weather radar viewpoint, of particular interest are those discussing the issue of contamination of weather radar data (Hood et al., 2010; Angulo et al., 2015; Lepetit et al., 2019); for other sectors, the

identification of adverse effects of wind turbines on the performance of air surveillance and marine radars is of great concern (Angulo et al., 2014; Cuadra et al., 2019). In general, wind turbine clutter reflectivity depends on various parameters such as wind turbine dimensions, incidence angle of radiation, rotor speed, blade pitch angles, nacelle orientation and radiation frequency (Gallardo-Hernando et al., 2011; Norin, 2015; Lainer et al., 2021). In literature, several papers about the reflectivity (and equivalent backscattering radar cross section) of the wind turbines can be found. One can separate the studies in those

dealing primarily with measurements (e.g., Bredemeyer et al., 2019; Kong et al., 2011; Kent et al., 2008) and others using numerical investigations of virtual wind turbine models (e.g., Muñoz-Ferreras et al., 2016; de la Vega et al., 2016).

On the contrary, research dealing with other polarimetric signatures of the WT is rare. In a recent (March 4-21, 2020), unique stare mode campaign held in Schaffhausen (Lainer et al., 2021), the WT is continuously illuminated by a fixed-pointing antenna; as emphasized by Reviewer 1 (Interactive comment on *Atmos. Meas. Tech. Discuss*, 2020;

https://amt.copernicus.org/preprints/amt-2020-384/amt-2020-384-RC1.pdf ): "the measurements as they are described provide further information on the properties of other polarimetric variables at the WT location. This information is urgently needed to comprehend the WT problem and I want to encourage the authors to add further publications based on this experiment". This preliminary study represents a small step in the direction of filling such polarimetric gap. However, it is important to point out that our main objective is an investigation of the dual-polarization backscattered signals by a wind turbine (WT)

when its rotor speed is very small or even close to zero, as well as during the transition from zero rotor speed to the ordinary moving conditions.

The reason is connected to the emerging interest toward "Bright" Scatterers (BS) (Rinehart, 1978) as additional tool for monitoring modern dual-polarization weather radars (Gabella, 2018). Thanks to the increased number of dual-polarization radars and in computational power for modeling and statistical analysis, a novel point of view regarding specific ground clutter

has emerged. It is no longer considered exclusively a disturbance that needs to be rejected; rather, its spatio-temporal properties are statistically characterized in order to be used for monitoring radar hardware. This is the case of the BS, which is a tall target, close to the radar and hit by the antenna beam axis. Recently, it has been shown that the "historical" polarimetric and spectral signatures of a BS in Switzerland represent a benchmark for an in-depth comparison after hardware replacements (Gabella, 2021). However, since it is hit during the operational weather scan program (Germann et al., 2015), the typical return

period for BS observations is as large as 5 minutes (300 s). Thanks to the recent unique MeteoSwiss stare mode campaign in Schaffhausen (March 4-21, 2020), the WT is continuously illuminated by a fixed-pointing antenna with a large number of pulses ($N = 128$). Using a PRF as large as 2000 Hz, dual-polarization signatures are available every 64 ms (128/2000). The





fixed pointing antenna turns out to be an important advantage if one aims at characterizing the intrinsic spectral signatures of the large, "bright" target.

A description of the radar, its scan program, the geographical area where it has been operating, and the observed WT is given in Sec. 2.1, while Sec. 2.2 presents the WT metadata, which are unfortunately available only every 600 s.

Sec. 3 represents the core of this manuscript: it will show that thanks to the very high temporal resolution, it is possible to give affirmative answers to the main questions that have stimulated the present study: does a still WT (rotor speed equal to 0) show stable and peculiar polarimetric signatures? Are they similar to those of a BS? Yes, indeed. The copolar coefficient is very

stable and close to 1; the dispersion of both the differential phase shift and the differential reflectivity is small; reflectivity values for both polarizations are stable (Sec. 3.1, from 17:00 to 17:10 UTC, zero rotor speed) … However, Sec. 3.2 will show that the situation becomes completely different when even a very small movement (and/or change in the aspect) takes place: small changes of the blade angles, small rotations of the nacelle and/or the rotor are able to cause significant changes in the polarimetric signatures even at the current 64 ms temporal resolution (from 17:10 to 17:20 UTC). Interestingly, Sec. 3.3 and

3.4 will show that the maxima constructive and destructive interference do not occur in ordinary moving condition (Sec. 3.3, stationary rotations for most of the ten minutes), rather during the small partial, "discontinuous" rotation that took place in the successive ten minutes (Sec. 3.4, a partial rotation of 216˚ sometime inside the temporal interval between 17:30 and 17:40 UTC). A thorough discussion is presented in Sec. 4; conclusions and outlook are found in Sec. 5.

## 2 Brief description of the experimental area, instrumentation and high temporal resolution data

### 2.1 The radar site (good visibility towards the wind turbine), observation geometry and the simple scan strategy

A dual-polarization, Doppler, mobile X-band radar has been used for the measurement campaign. A detailed technical overview of the radar system can be found in Neely et al. (2018). Some key specifications are listed in Table 2, page 3543 of Lainer et al, 2021. The radar site was near the city of Schaffhausen (approximately coordinates: 284 km North and 692 km East, using the Swiss LV03 conformal reference system), at an altitude of 455 m. The three wind turbines of the small wind

park located north of Schaffhausen are installed on a hill surrounded by forests. For the specifications and other properties (including geometry) of the wind turbines, the reader may refer to Table 1 in Lainer et al. (2021). For the whole campaign in 2020, we observed only the wind turbine with the best visibility and maximum reflectivity value observed during the 2019 campaign: in the paper by Lainer et al. (2021), it is labelled as WT1, here after it will be simply labeled as WT. The horizontal distance from its mast and the radar site is 7.76 km. By analyzing the output of the simulations by the X-band Ground Echo

Clutter Simulator (GECS-X) described by Gabella et al. (2008), which has been run using a digital elevation model (DEM) with 50m resolution, the radar visibility towards the wind turbines could be determined. The used approach follows the simple but effective geometric optics assumption described in Gabella and Perona (1998). From the "visibility" map (see Fig. 1a in Lainer et al., 2021), one gets the minimum angle of elevation at which a target could be "seen" from the radar site, which is 2.25˚. If no obstacles were present on the surface, then the base of the WT at ~765 m would be visible from the radar site: the



nominal angle of elevation using simple trigonometry (and flat Earth) turns to be, in fact, 2.305˚ (at range 7766 m). A wood of conifers is instead present between the radar and the WT: those tall trees, at approximately 1km range, partially block just a small part of the main lobe towards the base of the mast; on the contrary, the rotor center of the WT is always visible: knowing the nacelle height, in fact, it is easy to derive that the angle of elevation of the rotor center is 3.308˚ (range is ~7773 m). Finally, the angle of elevation for the vertical pointing end of a blade is 3.789 ˚ (range is ~7777 m).

For the peculiar stare-mode strategy of the March 2020 campaign, we have opted for an angle of elevation of 3.1˚: consequently, the whole half power beam width (HPBW, from 2.45 ˚ to 3.75˚, in the elevation plane) is practically not subject to occultation by obstacles. The azimuth was set to 338.9˚. In this study we will present polarimetric signatures derived using I and Q data of Gate103 (starting from 0), which ranges from 7725 m to 7800 m. At such range, the size of the pencil beam HPBW is about 180 m. On the contrary, the range resolution is independent of range: being a priori known at what range the

(weather) target should be detected and investigated, it can be pushed down to half the pulse width multiplied by the speed of light. This is in fact the case for our X-band radar with a pulse width of 500 ns (specifications and more details regarding the radar can be found in Table 2 of Lainer et al., 2021). It is then clear that the WT target is thoroughly bounded inside the radar sampling volume of 180×180×75 m (0.0243 km³) only as long as the nacelle orientation is around 0˚ or 180˚. When the orientation goes toward 90˚ or 270˚, part of the 65 m blades (130 m diameter) will exceed the range resolution. It is also evident

that the incident electromagnetic field transmitted by the radar is far from being planar over the extent of the target nor the WT can be assumed to be a point target, in order to retrieve a value of radar cross section from the measured power, in turn converted into radar reflectivity using the Probert-Jones (1962) approximation (Gaussian distribution of the radiated power over the main lobe). If one pretended the point target radar equation (see, e.g., eq. (1) in Lainer et al., 2021) being applicable and compared it with the meteorological radar equation (see, e.g., eq. (6) in Lainer et al., 2021), then the RCS (in dB square

meters) could be derived by simply decreasing by 34.4 dB the radar reflectivity factor (expressed in dBz, see Sec. 2.3.1).

**2.2 Wind turbine data and metadata collection: a very peculiar 40-min interval under detailed investigation.**

The focus of the present study is limited to dual-polarization backscattered signals in correspondence of a situation with zero rotor speed. Hence, the prerequisite is the presence of a 10 min interval without any rotor rotation. Despite being unusual, this situation has anyhow happened on the very first day of the campaign, namely between 17:00 and 17:10 UTC on March 4,

2020. We are aware of such special conditions thanks to Hegauwind GmbH & Co. KG Verenafohren that have kindly provided the operational data of the wind turbines. These include environmental (wind speed, direction, outside Temperature, …), instrumental (indoor and hardware Temperature, current, voltage, power) and operational (nacelle direction, rotor speed, pitch angle of the three blades, …) for a total of almost a hundred parameters. Unfortunately, such abundance of parameters cannot compensate the main limitation of these data if compared with the very high-temporal resolution of radar echoes, which are

available every 64 ms: the granularity of the available wind turbine data; we have to cope with a very poor 10-min temporal resolution, to be somehow associated to 9375 radar echoes that are available every 10 minutes. As shown by Lainer et al. (2021), the average rotor speed and the pitch angle of the three blades are by far the most important information for radar-





related studies. For instance, zero (or very small) rotor speed is typically associated to a large value of the angle of the 3 blades, as it can be seen in Fig. 1 (red vs blue dots).

During the first 12 hour of the campaign (March 4, 2020), 10-min average rotor speed was particularly large, ranging from 7 to 11 rpm (blade angles close to 0). But in the second part of the day, which is displayed in Fig. 1, a 4-hour period with an almost constant and regular rotor speed of about 7 rpm has taken place, followed by a quiet period that was approaching in the last twenty minutes preceding 17 UTC (average rotor speed around 0.1 rpm, red dots; blade angles at 70˚, blue dots). In particular, the conditions during the 10 minutes after 17 UTC on March 4, 2020 were ideal from our viewpoint: the average

rotor speed was exactly 0 rpm, so that we know no movements have happened (the blade angles were also kept constant at 70˚). Unimportant if during almost 8 minutes no stare mode radar data are available: during the two final minutes with available radar data, we will see that radar measurables are very stable with no (or very little ) variability. This fact will be investigated and shown at the original very high temporal resolution of 64 ms and displayed using a re-sampled 8 s temporal resolution in Sec. 3.1. The (8 s) low resolution analysis is based on the Maximum, minimum, mode and median values of 125 original (64

ms) echoes. The nacelle orientation with respect to the radar beam axis was about 61˚.

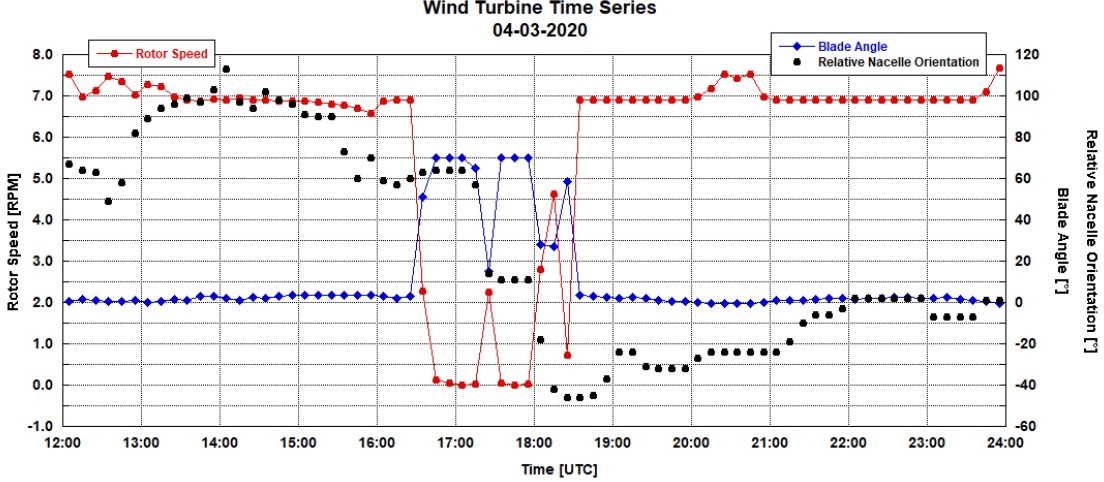

**Figure 1.** Wind turbine blade angles and rotor speed on March 4, 2020.

In the successive 10-min interval, namely between 17:10 and 17:20 UTC, the rotor has turned by 0.2 rotation, which is 72˚: analyses will be presented in Sec. 3.2. In this 10-min interval, the blade angles have been reduced from 70˚ to 65˚, while the nacelle orientation has changed only by a few degrees: from 61˚ to 57˚ (see Fig. 1, black dots, *y-axis* on the right).

Then, between 17:20 and 17:30, the rotor has started its typical rotation, despite at a speed smaller than usual (2.25 rpm) with blade angles a bit larger than usual, but still close to just a few degrees. The nacelle orientation has changed significantly: from

57˚ to just a few degrees, where it remains also for the next 10 minutes, from 17:30 to 17:40 UTC. This is again a 10-min interval with a small partial rotation: just 216˚, with a completely different value of the blade angles, which have been again



set to 70˚ (same value as from 17:00 to 17:10 UTC). Interestingly, the largest RCS value at horizontal polarization has occurred twice (17:31:29 and 17:35:53 UTC) with this configuration (see Sec 4 for more details).

## 2.3 The polarimetric weather radar measurables (available every 64 ms)

**2.3.1 First measurable: radar reflectivity at horizontal and vertical polarization**

The first (and probably most known) meteorological quantity measured by weather radar in the history of radar meteorology is the so-called radar reflectivity. The backscattered received power, $p_r$, caused by the hydrometeors and detected by the radar is, in fact, directly proportional to the radar reflectivity, z. Since both the received power and the radar reflectivity span several order of magnitude, they are often expressed using a Log-transformed scale, after having divided the physical quantity by a

normalization factor. For linear power, $p_r$, the normalization value is typically, $p_0 = 1$ mW. The typical normalization value for the reflectivity is $z_0 = 1$ mm$^6$/m$^3$. The dual-polarization radar can simultaneously measure two reflectivity values that are *de facto* orthogonal: they will be indicated as $z_h$ and $z_v$ in linear units or $Z_h$ and $Z_v$ after the Log-transformation. As stated, [zh] = [zv] = mm$^6$/m$^3$, while [$Z_h$] = [$Z_v$] = dBz. The upper case $P_r$ indicate the Log-transformed received power, where [$P_r$] = dBm. As far as the quantization is concerned, a value of 0.5 dBz has been chosen by the radar manufacturer. At MeteoSwiss, an

identical choice has been done regarding the reflectivity resolution of the five C-band radars of the Swiss network; also the formula from converting from 8 bits to physical value is identical. The linear conversion from DN to Log-transformed radar reflectivity is the following:

$$Z_{dBz} = (DN-64)/2. \tag{1}$$

However, the maximum recorded value observed in the practice in Switzerland (both in the C-band network and with this

mobile X-band radar) rarely exceeds 85 dBZ (DN=234); furthermore, a weak echo corresponding to DN=14 (-25 dBZ) can, only be detected at a range of 1 km or closer from the X-band radar.

## 2.3.2 Second measurable: differential reflectivity

The differential radar reflectivity, Zdr, is an important polarimetric quantity that can be derived by combining the previously described two measurables in a differential manner: it is defined as the Log-transformed ratio between the copolar linear

reflectivity measured using horizontal ($z_h$) and vertical ($z_v$) polarizations. In formulas:

$$Z_{dr} = 10 \, Log \, \left( z_{h/} z_v \right). \tag{2}$$

The differential reflectivity is expressed in dB, and a value of 0 dB means that $z_h = z_v$. In practice, $Z_{dr}$ can also be computed as the difference between Zh and Zv. The differential reflectivity was introduced by Seliga and Bringi [23] for a better estimate of rainfall since it contributes to reducing the uncertainty associated with raindrop size distributions. Indeed, the information

associated with $Z_{dr}$ is remarkable; however, the issue of a proper calibration remains a challenge for successful quantitative





precipitation estimation. As far as the quantization is concerned, 256 values (8 bits) are linearly assigned by the manufacturer over an interval that spans 16 dB (from -8 to + 8 dB). We will see in Sec. 3 that that, surprisingly, many WT echoes are outside this interval. Consequently, in this study, we abandon such 1/16 dB radiometric resolution and use the poorer 0.5 dB resolution that permits us to derive the value in all circumstances simply as the difference between $Z_H$ and $Z_V$.

### 2.3.3 Third measurable: module of the copolar correlation coefficient between horizontal and vertical polarization

A very important quantity measured by dual-polarization radars is the correlation between the copolar horizontal, HH, and vertical VV returns, called the copolar correlation coefficient (often referred to as $\rho_{HV}$ or $\rho_{co}$). It is worth noting that the copolar correlation coefficient is directly connected with the differential reflectivity: it can, in fact, be seen as a measure of the dispersion of the differential reflectivity of the 128 instantaneous backscattered signals (with pulse repetition time of 0.5 ms) used to derive each echo, obviously every 64 ms. For a detailed and clear description of the interesting and complicated nature of this measurable, the reader may refer to the electronic supplement (e06.1) accompanying the book by Fabry [22]. Here it is sufficient to remind that, being the module of the complex correlation coefficient between two orthogonal components (represented by two complex numbers) of the backscattered electromagnetic field within the radar sampling volume, it ranges from 0 (no correlation between the two polarizations) to 1 (perfect correlation). If targets within the radar sampling volume were similar, then the time series of signals at horizontal and vertical polarizations would be highly correlated both in amplitude and phase. On the contrary, the greater the variability in shapes of the targets, the smaller will be the value of $\rho_{HV}$. When many backscatterers are randomly distributed within the backscattering sampling volume, the copolar correlation coefficient is considered a measurement of shape diversity. Consequently, the echoes of light rain and drizzle (small and similar spherical drops) are associated with very large values of $\rho_{HV}$, mostly larger than 0.995; $\rho_{HV}$ values in melting snow are lower (typically between 0.8 and 0.9) and make the melting layer easily distinguishable. If the sampling volume contains a significant number of different targets, such as what happens with ground clutter, $\rho_{HV}$ will decrease considerably. In particular, the range of $\rho_{HV}$ for most ground clutter echoes is between 0.650 and 0.950. Since the most interesting values are very often close to 1, typically a Logarithmic function is used in the quantization process when assigning a DN to the original floating point value of $\rho_{HV}$. (see for instance eq. 6 in Gabella (2018) for the MeteoSwiss quantization formula that permits increments as small as 0.0001 when close to 1). The X-band radar manufacturer has opted for a linear stretch from 0 to 1, which means equal increments of 0.0039 over the whole interval. This choice is certainly not optimal for the present study, which focuses on the similarity between the polarimetric signatures of a single radar bin that contains a still WT compared to a single bin that contains a BS. For instance, for the 1440 echoes (5 clear sky days) of the Cimetta BS (see Sec. 3.5.1 of Gabella, 2018), the present quantization would use only use 4 DNs (from 252 to 255, with mode at DN = 254, which is $\rho_{HV} = 0.9961$), while MeteoSwiss quantization had DNs ranging from 180 to 251, with mode at DN=233, which is $\rho_{HV} = 0.9982$. The median (standard deviation) would be 0.9961 (0.0028), instead of 0.9968 (0.0024). Both quantization choices are unbiased, the average value is 0.9962.



### 2.3.4 Fourth measurable: differential phase shift

Another polarimetric quantity measured by the dual-polarization radar is the differential phase shift, $\Psi_{dp}$, between the phase of the copolar signal at horizontal and vertical polarization, respectively. Apart from an arbitrary offset value $\Psi_0$, which can

be compensated via software, such difference between the phase of the two orthogonal polarizations arises from two effects:

- A difference in the delay introduced by the scattering of the transmitted wave, known as the backscattering phase shift, $\delta_{co}$.

- A difference in the forward propagation velocity of the two polarizations (reaching the target and coming back to the radar), known as the differential propagation phase, $\Phi_{dp}$.

Keeping in mind $\Psi_0$, as well as the two important above mentioned terms, then the differential phase delay, $\Psi_{dp}$, at any given range, $r$, can be described using a simple formula:

$$\Psi_{dp} = \delta_{co} + \Phi_{dp} + \Psi_0, \qquad (3)$$

which is the sum

1. Of the backscattering phase delay of the target at that specific range, $r$.

2. Of the two-way differential propagation phase that occurred when propagating from the radar to the observed target and then back to the radar.

3. Of an offset value $\Psi_0$, which is the phase difference between the two transmitted polarizations at range zero.

At the beginning of the Schaffhausen campaign, the constant $\Psi_0$, which depends on the radar hardware components and design, has been set to a small positive value close to zero. During dry days, like March 4, 2020, also $\Phi_{dp}$ does not vary and can be

assumed to be zero. Hence, what is observed when analyzing the dispersion of $\Psi_{dp}$, is basically the dispersion of the differential backscattering phase delay, $\delta_{co}$. For most ground clutter targets, the dispersion is very large, being its distribution close to a uniform distribution (in this case a standard deviation of $360°/120^{0.5}$ would be expected). On the contrary, in the case of a Bright Scatter (e.g., the tall Cimetta tower presented in Gabella, 2018), the dispersion is small: for instance, the daily standard deviation (288 echoes) of $\Psi_{dp}$ was ~4° in four (out of 5) days analyzed (see Section 3.6 in Gabella, 2018). Something similar

could be assumed for a perfectly still WT (zero rotor speed, no changes in nacelle orientation nor in blade angles): as a matter of fact, on March 4, from 17:13:58 to 17:14:18 UTC, the standard deviation of 313 $\Psi_{dp}$ values is as small as 3.1˚ (see Sec. 3.2). Similarly, from 17:14:33 to 17:14:41 UTC, the standard deviation of 313 $\Psi_{dp}$ values is 3.6˚ (see also the last 8 s in the figure shown in Sec. 3.2).

### 3 Main results using a 8 s temporal resolution for visualization purposes: from 17:08 UTC to 17:40 UTC

We will show in this descriptive Sec. 3 that for the purpose of visualization and analysis, a small set of four statistical values derived (and displayed) every 8 s and from the original 64 ms echoes, is adequate and satisfactory in order to try to characterize





the backscattering properties of the WT. Two of these values represent the central and most probable locations of the original 125 echoes available every 8 s: the median and the mode. The other two descriptors delimit the extreme boundaries of the 125 echoes: the maximum and the minimum.

For a WT, a situation without any movement of the rotor is certainly not a usual one. However, as described in Sec. 2.2, this interesting configuration has already took place during the first day of the 3-week campaign, namely between 17:00 and 17:10 UTC (March 4, 2020). This can be seen in Fig. 2, which shows, at the upscaled 8 s resolution, the median (blue curve), mode (green), MAX. (red) and minimum (cyan) radar reflectivity values for the horizontal polarization. (As already mentioned in Sec. 2, radar echoes are only available starting from 17:08 UTC, not at 17:00 UTC.). All the four descriptors are coincident

until approximately 17:11 UTC. On the one hand, the backscattered power at horizontal polarization is characterized by an amazing stability ($Z_H$ persistently equal to 56.5 dBz, which is variability smaller than ±0.25 dBz). On the other hand, we dare extrapolating a similar (if not identical) value for the first 8 minutes, during which the radar was performing the PPI scan centered around 17 UTC. A similar situation has also characterized the copolar correlation coefficient, which has always been equal to 1 (8 bits always set to 1, namely DN=255), as it can be seen in Fig. 3. The four descriptors are coincident until almost

17:14 UTC. This means that $\rho_{HV}$ had same DN for more 5500 consecutive echoes. More details regarding polarimetric signatures corresponding to WT zero rotor speed are presented in Sec. 3.1 (WT metadata that refer to the 17:00-17:10 UTC interval), while those related to small movements of the successive ten minutes are described in Sec. 3.2.

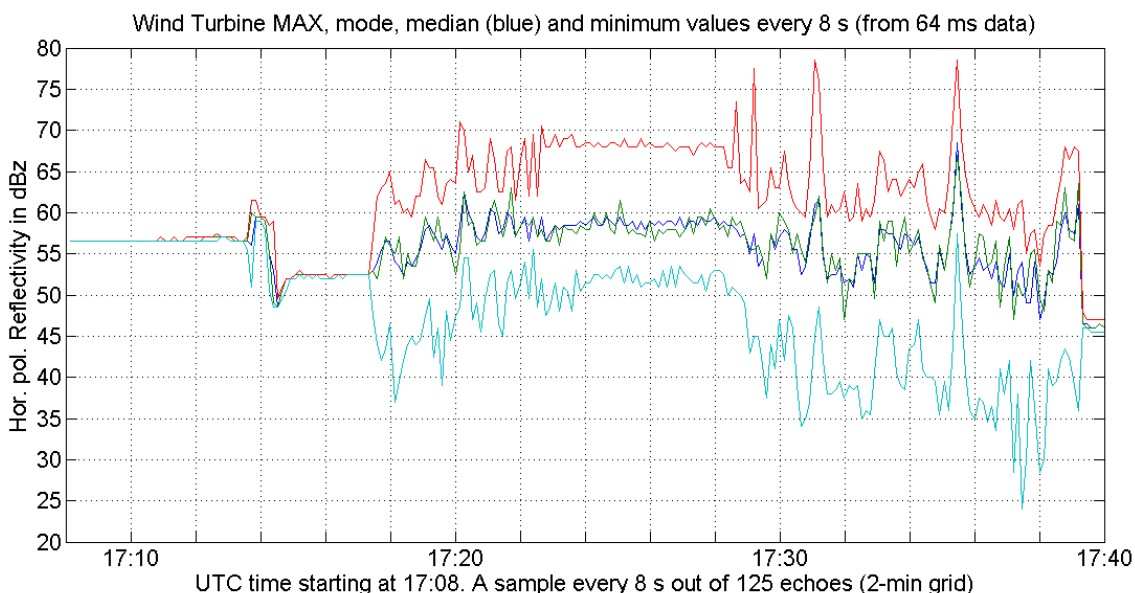

**Figure 2.** Time line plot of horizontal polarization reflectivity for the 75 m radar gate that contains the pole of the wind turbine. The solid lines join 8 s statistical values (median using blue, mode using green, MAX using red, min. using cyan) obtained by using 125 consecutive radar echoes at the original 64 ms resolution. (If all the 125 echoes have the same values, only the last color used is visible, namely cyan). Being the visualization based on 8 s points, the solid lines consist of 240 points that cover 32 minutes (15 points every 2 minutes, which is in correspondence of the vertical grid lines).

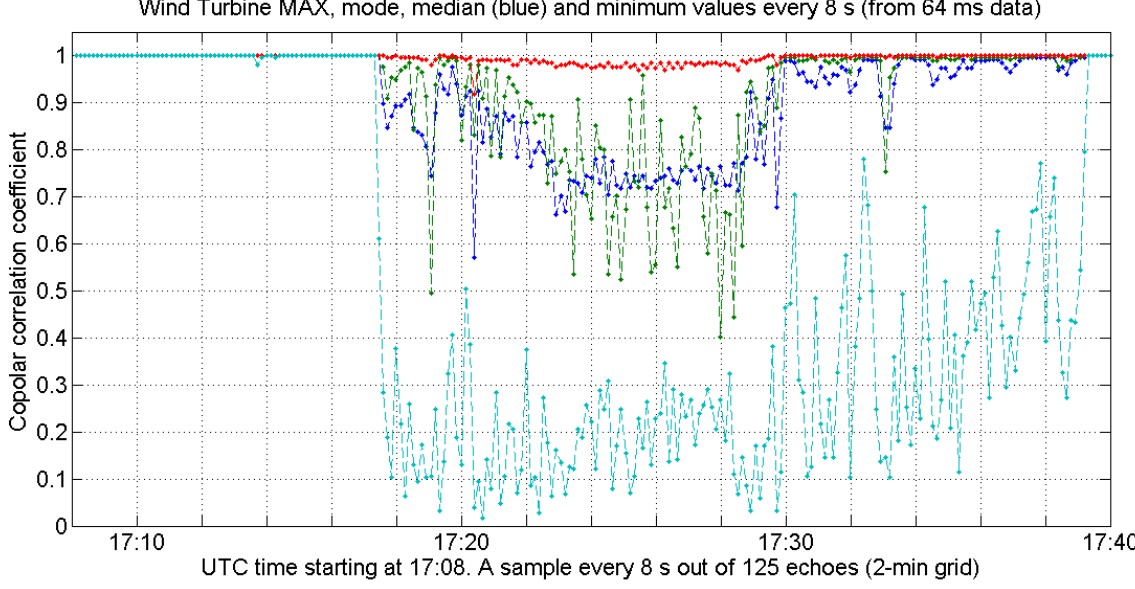


**Figure 3.** Time line plot of the copolar correlation coefficient for the 75 m radar gate that contains the pole of the wind turbine. The dashed lines join 8 s statistical values (shown by mean of a small cross) obtained by using (for each point) 125 (different) consecutive echoes at the original 64 ms resolution. Same color code as in Fig. 2. It is interesting to note that the "0 rotor speed condition" read from the WT metadata for the 17:00-17:10 and 17:40-17:50 ten-minute intervals, is probably prolonged also for several minutes after 17:10 as well as anticipated

for ~40 s  before 17:40 UTC.

### 3.1 From 17:08 to 17:10 UTC: 2 min of stare mode radar data (1875 echoes) corresponding to 0 rotor speed

As already described in the introductory part of Sec. 3, in correspondence of zero rotor speed, two polarimetric signatures were perfectly constant: the reflectivity at horizontal polarization ($Z_H$ = 56.5 dBz) and the copolar correlation coefficient ($\rho_{HV}$ = 1).

What about the reflectivity at vertical polarization? Although not perfectly constant, it was bounded between 38.5 dBz and 41.5 dBz, as it can be observed in Fig. 4; the mode occurs at 39.0 dBz, the median (mean) value is 40.0 (39.9) dBz. It is interesting to note that at the original (very high) temporal resolution of 64 ms, all the reflectivity changes from one echo to the next one, were either 0 dBz or ±0.5dBz.

For these two minutes, the curve of differential reflectivity (Fig. 5) is of no particular interest, being simply $Z_V$ after a change

of sign, plus the constant value of $Z_H$. On the contrary, it is interesting and impressive how far the differential reflectivity is from a "neutral" interval (centred around 0 dB): from the above written figures, it is straightforward to derive that median value of $Z_{DR}$ is as large as 16.0 dB. It could be caused by the specific stop positions of the rotor, combined with pitch angle of the blades. While in general, when the blades are rotating (rotor speed above 1 rpm), one could expect median (and mode) values not too far from 0 dB. This is in fact the case between 17:20 and 17:30 UTC, see Fig. 5 and the thorough description in

Sec. 3.3.

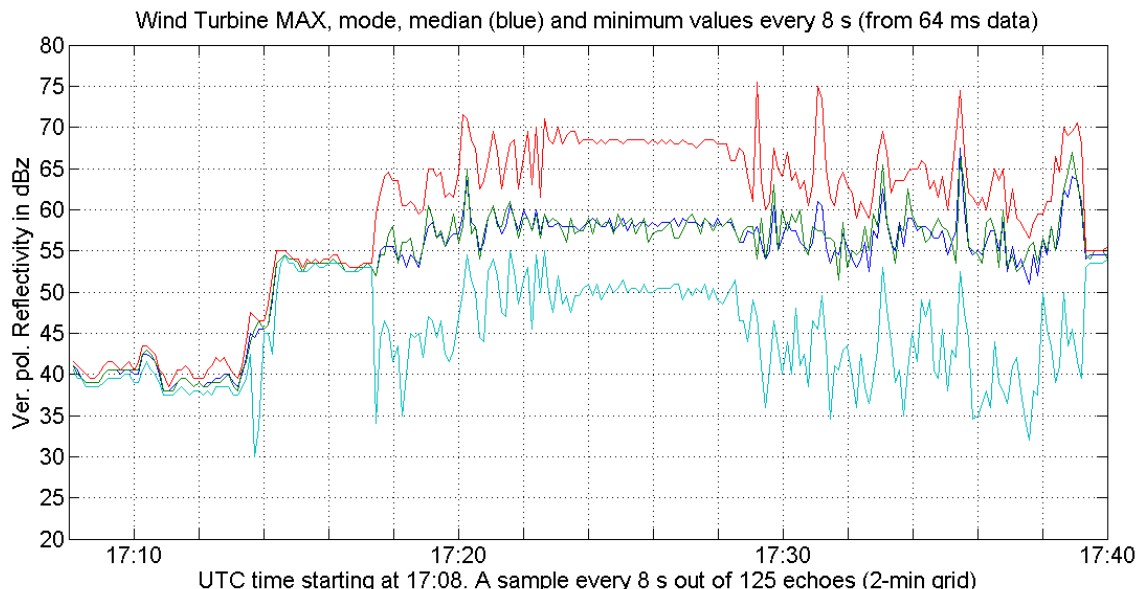

**Figure 4.** Same as Fig. 2, but for the vertical polarization.

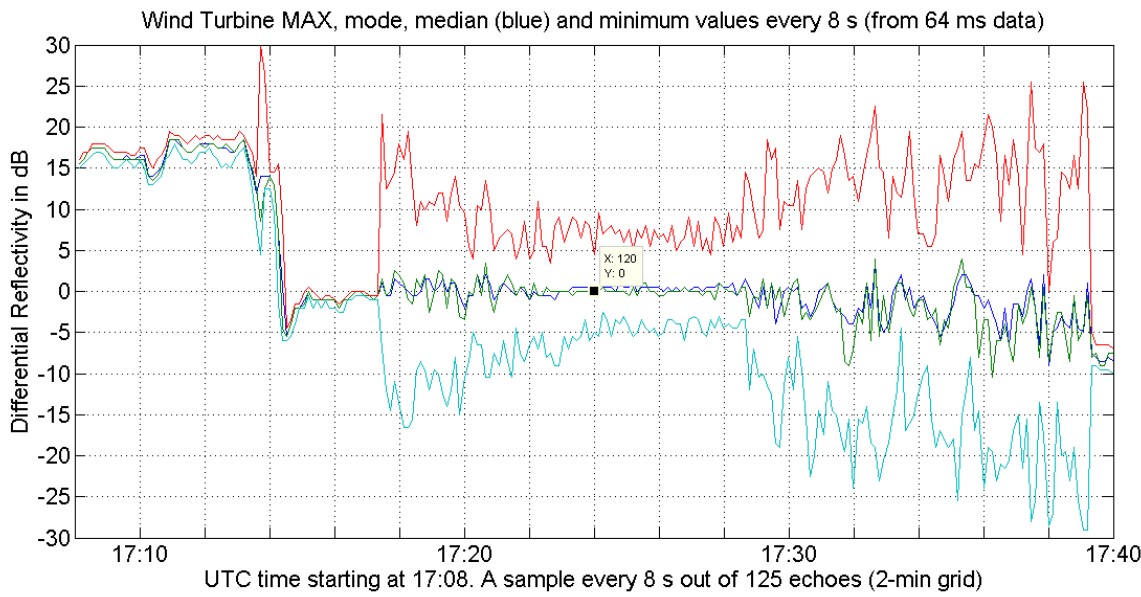

**Figure 5.** Same as Fig. 2, but for the dimensionless differential reflectivity. Being the visualization based on 8 s points, the solid lines consist of 240 points that cover 32 minutes (15 points every 2 minutes, the vertical grid dashed lines). Most of the 125 echoes between 17:24:00 and 17:24:08 recorded a differential reflectivity value equal to 0 dB ($Z_H = Z_V$). During this 8 s, $Z_H$ ($Z_V$) has never exceeded $Z_V$ ($Z_H$) by more than 5 dB (64 ms sampling time, which means 125 echoes). Although the median and the mode are in general around 0 dB, they are not during the intial and final intervals characterized by 0 rotor speed: our hypothesis is that in steady condition the differential reflectivity depends on the position of the blades as well as their orientation angles.



In case of zero rotor speed, another statistical parameter of particular interest is the dispersion of the differential phase shift, which in the absence of precipitation, was, in fact, coincident with the dispersion of the differential backscattering phase shift, $\delta_{co}$. As shown by Gabella (2021, 2018), very large median values as well as very small standard deviation of $\rho_{HV}$ are typical of Bright Scatterers. For randomly distributed Rayleigh backscattering targets, a standard deviation of 360° divided by the square root of 12 would be expected, namely 103.9°. On the contrary, for the steady WT a much smaller value is expected, ideally just a couple of degrees. This is in fact the case: the standard deviation of these 2-minute values is as small as 3.0°.

### 3.2 From 17:10 to 17:20 UTC: blade pitch angle changed from 70° to 65° and small partial rotation

The successive ten minutes are also characterized by quasi-null rotor speed: the average value of $r_s$ is as small as 0.02 rpm between 17:10 and 17:20 UTC. This means that only a partial rotation of 72° has occurred during the whole 10 minutes. A careful analysis of the copolar correlation coefficient (Fig. 3) shows that:

- Some short-lived event has happened just before and after 17:14 UTC (a few 64 ms $\rho_{HV}$ echoes just smaller than 1);
- Something has started just before 17:17:40, causing a remarkable decrease of $\rho_{HV}$ until 17:30.

The changes that take place before and after 17:14 UTC have huge impact on the vertical reflectivity and on the differential backscattering phase shift (and, in turn, $\Psi_{dp}$, see eq. 3). The copolar correlation coefficient, $\rho_{HV}$, is different from 1 (DN=255) in just 22 cases (64 ms echoes) that belong to three extremely short periods that are inside three different 8 s interval, as it can be seen in Fig. 3. The smallest value of $\rho_{HV}$ is 0.9803 (DN=250) that just happens once. Also the $2^{nd}$ minimum (DN=252) happens once, while there are two 64 ms echoes with DN=252. In particular, these 4 smallest $\rho_{HV}$ values are consecutive in time and perfectly correspond to the 4 smallest value of vertical reflectivity (with an absolute minimum of 30 dBz, which can be easily identified in Fig. 4 but also in Fig .5, hence deducing that the corresponding value of $Z_H$ was 60 dBz). While the drops of $Z_V$ consist at "high frequency" ($1/0.064$ $s^{-1}$ = 15.625 Hz) of "up-down" jumps (hence, the first temporal derivative changes very often its sign), the changes in the differential phase shift, $\Psi_{dp}$, are characterized by long sequences of ("high frequency") negative discrete derivative values which bring $\Psi_{dp}$ from the original ("equilibrium") value not far from 0° down to the new ("equilibrium") value of -466°. This fact can be seen in Fig. 6, which shows the large changes in the differential phase shift: the original aliased values (between ±180°) detected by the radar are shown using blue dots, while the more meaningful de-aliased curve is shown using a solid red line that links all the points.

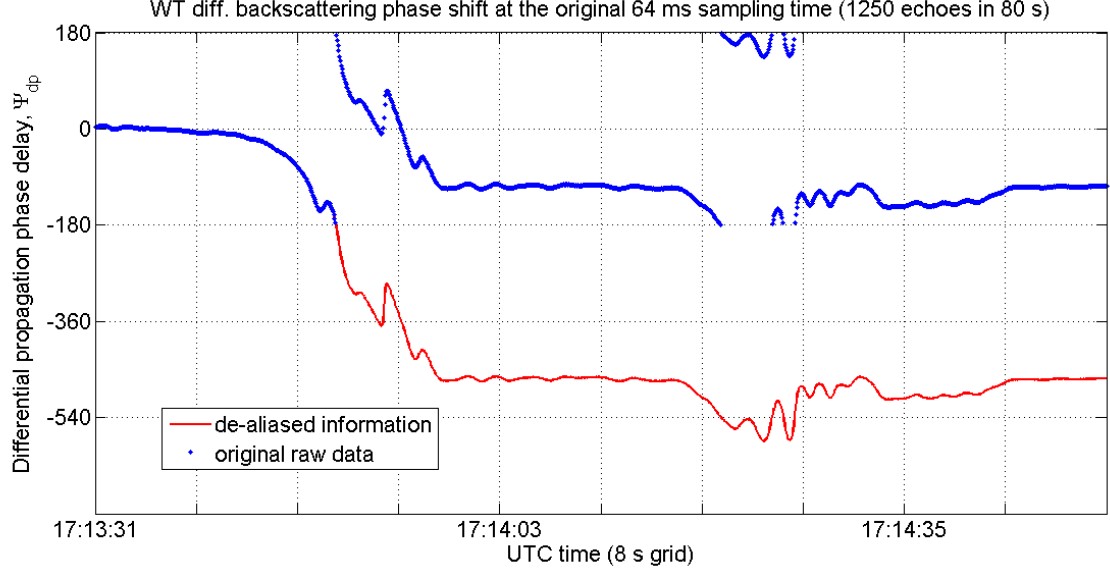

**Figure 6.** Representation of the variability of the differential phase shift measurable (see Sec. 2.3.4) at the highest available temporal resolution, namely 64 ms. The abscissa spans an interval of exactly 80 s; vertical lines on the $x$-axis are every 8 s, which corresponds exactly to 125 echoes. Every 64-ms radar echo (measurable) has been derived by means of 128 pulses transmitted using a Pulse Repetition Frequency of 2000 Hz. The blue dots corresponds to the raw (aliased) data, while the red lines shows the proper evolution of the signal. Note that being the radar receiver stable in phase and amplitude and clear sky conditions, changes in the differential phase shift, $\Psi_{dp}$, can be attributed to changes in the differential backscattering coefficient, $\delta_{co}$.

Between 17:12:59 and 17:13:31, which is the starting time of the high-frequency values shown in Fig. 6, $\Psi_{dp}$ was bounded between 11° and +5°; during the first 200 echoes (12.8 s) of the present interval, $\Psi_{dp}$ has already decreased to approximately -20°. Then the slope of the decay starts to increase further and further until a first relative minimum of -369° is reached at echo #355 (the slope has decreased to 0, obviously). Exactly in correspondence of the first, "longer" (9 consecutive 64 ms echoes) and deeper (down to 0.9803, which is DN=250) drop of $\rho_{HV}$, $\Psi_{dp}$ starts to increase again up to -290°. Then, a rapid decrease down to -466° follows, which is reached around echo #425 (17:13:58.2 s). Obviously, in the original, aliased data delivered by the radar signal processor, -466° corresponds to a value of -106°. Except a few oscillations between 17:14:13 and 17:14:35, the new "equilibrium" value is kept until 17:17:37.4 s, when a new remarkable change will start.

Adding up, between approximately 17:13:31 and 17:14:41, something have caused:

- (large) changes in $(Z_V)$ $Z_H$ that combined cause an extreme variability of $Z_{dr}$, with a maximum of + 30 dB;
- the consequent $Z_{dr}$ transition from an unexpectedly very large value of ~16 dB to an "expected" value close to 0 dB;
- the transition of $\Psi_{dp}$ (actually, of the differential backscattering phase shift, $\delta_{co}$) from ~0° to -466°.

Is it related to the change in the blades angle from 70° to 65°? Or is it related to the change of the nacelle orientation with respect to the radar from 61° to 57°? Or both? Indeed, one important limitation of the present analysis is due to the very low





temporal resolution (sampling time equal to 600 s) of the ancillary data associated to the WT. For instance, what caused (between 17:14:13 and 17:14:45) the further decrease of $\Psi_{dp}$ down to less than -540° and then back to -466°?

Finally, during which part of the 10-min interval has the 72°-rotation of the rotor took place? It seems reasonable to think that such rotation has started around 17:17 UTC, as it could be argued by differences between the 8-s maximum (red) and minimum (cyan) in Fig. 2 ($Z_H$), Fig. 4 ($Z_V$), and Fig. 4 ($Z_{dr}$). If one were interested to determine with more precision the starting time, he could use the (15.625 Hz) "high-frequency" $\rho_{HV}$ echoes: the constant position of 1 (DN=255) is abandoned exactly at 17:17:17 UTC plus 366 ms. Then $\rho_{HV}$ is characterized by a large dispersion until 20 s before 17:30 UTC, when the rotor speed again

slows down considerably and blade angles goes back to 70° (see next Sections 3.3 and 3.4).

### 3.3 From 17:20 to 17:30 UTC: 22.5 rotor revolutions, blade pitch angle changed from 65° to 15°

From 17:20 and 17:30 UTC, the average 10-min value of $r_s$ is 2.25 rpm, which implies 22.5 revolutions in such 10-min interval. As far as the blade angle is concerned, it has decreased from 65° to 15°. The whole 10-min interval is characterized by heavy fluctuations of $\rho_{HV}$, which never reach anymore the value of 1 (red curve in Fig. 3); every 8 s, the mode (blue) and median (green)

values, which have been derived using 125 echoes, are very different. Regarding the fluctuations of the maximum and minimum reflectivity values of both polarizations (red and cyan curves in Fig. 2 and 4), they are smaller between 17:23 and 17:28 UTC; our hypothesis is that during these 5 minutes the rotation was faster than the 10- min average, while before 17:23 UTC and after 17:28 UTC, only a partial, slow rotation was occurring, similar to the one before 17:20 UTC. During this period with efficient rotor speed (say, 4-5 rpm) for energy production, both polarizations show median reflectivity values around 58

dBz; consequently, the median $Z_{dr}$ is around the "expected" value of 0 dB.

    It is particularly interesting that, while the rotor is probably slowing down (precisely at 17:29:31.729), $Z_H$ reaches the 3rd maximum value of the whole campaign, which is 77.5 dBz, in correspondence of $Z_{dr} = 7$ dB (the previous echo was 73.5 dBz, the following one 75.5 dBz). The 3rd maximum value of $Z_V$ can be identified 320 ms earlier (5 echoes back in time), in correspondence of $Z_{dr} = 1$ dB. During these 10 minutes the nacelle orientation changes from 57° to around 10°, where it will

remain also in the successive 10 minutes (see Sec. 3.4).

### 3.4 From 17:30 to 17:40 UTC: blade pitch angle back to 70° and again partial rotation

During the quasi-steady 17:30-17:40 interval, the average rotor speed was 0.06 rpm; this means that the rotor has turned overall only by 0.6 rotation, which is 216°. In Sec. 3.2 we had assumed that the partial rotation of 72° took place only after 17:17:17 UTC (and until 17:20 UTC); similarly, we here assume that the 216° degree rotation has been occurring (a few tens of seconds)

before 17:39:41 UTC, when the value of $\rho_{HV}$ became again persistently equal to 1 (see Fig. 3). It is worth noting that the absolute Maximum reflectivity value of the whole campaign (78.5 dBz) has been detected in four echoes at such very low rotor speed (0.06 rpm, on average, over the whole 10 minutes). The four echoes belong to two different 8 s interval and in both cases (two absolute peaks in the red curve inf Fig. 2) the 64 ms echoes are consecutive: the first pair is at 17:31:29.167 and





17:31:29.231 UTC, respectively (the corresponding $Z_{dr}$ values are 4.5 and 4.0 dB); the second pair is at 17:35.53.367 and
17:35:431 UTC, respectively (the corresponding $Z_{dr}$ values are 5.5 and 6.0 dB). The nacelle orientation is around 10°, which
is one (among several) 10-deg bin where the absolute maximum of 78.5 dBz has been recorded during the campaign; other
orientations involved are around 110°, 170°, 260° and 340°, as the interested reader can see in Fig. 10(a) of Lainer et al. (2021)
It is interesting to note that the slow rotation corresponds again to larger fluctuations of the maximum and minimum reflectivity
values of both polarizations (red and cyan curves in Fig. 2 and 4), as described in Sec. 3.3 for the first 3 minutes.

Around 17:39:20 UTC, the rotor probably stops its rotation ($\rho_{HV}$ often equal 1):

- $Z_V$ is bounded between 53.5 and 55.5 dBz; smaller than the median of the "energy production" WT mode (for instance,
  between 17:23 and 17:28 UTC see Sec. 3.3); much larger than 40 dBz, which is the median values of the previous 0
  rotor speed interval (see Sec. 3.1).

- $Z_H$ is bounded between 45.5 and 47.0 dBz, even smaller the median of the 5-min "energy production". However,
during the previous 0 rotor speed interval (see Sec. 3.1), $Z_H$ was surprisingly constant and equal to 56.5 dBz (see Sec.
  3.1), which is a value close to the median of the 5-min "energy production".

- Consequently, the median differential reflectivity of this 0 rotor speed interval is -8.0 dB, while in the one described
  in Sec. 3.1 it was at +16.0 dB!

## 4 Discussion

In this preliminary investigation, we have thoroughly analyzed 30 thousands polarimetric echoes acquired in 32 minutes during
which the WT rotor has accomplished 23.3 rotations. Thanks to the 10-min ancillary information regarding the WT, we know
that the rotor speed was exactly zero during the first 2 minutes. It is also very likely that rotor speed was zero during the last
40 s (from 17:31:40 to 17:32 UTC, see Figs. 2-5). If compared to its ordinary rotation conditions, a still WT is much easier to
be identified and rejected as clutter. This is something that has been known for a long time. The deep and detailed analysis
395   presented here shows something novel in view of the emerging interest in BS as additional source of information for monitoring
dual-polarization weather radars and meteorological applications (e.g., for assessing the path integrated attenuation of a melting
hail cell, see Gabella et al. 2021). Indeed, the polarimetric signatures of the present still WT are similar to those of a BS in
terms of very small dispersion (both copolar correlation coefficient, $\rho_{HV}$, and differential phase shift) and large average and
median values of $\rho_{HV}$. Actually, the dispersion is even smaller and the central value even closer to the unity asymptotic limit
400   than for the BS investigated at C-band with a rotating antenna in a previous study. We think that such fact is due to the special
stare mode antenna scan program: all the 128 averaged pulses refer to an antenna beam axis that is pointing in the same
geometrical direction. Residual sources of variability are then only fluctuations of the tropospheric refraction index and small
movements of the blade tips. Similarly, for a still WT, the dispersion of both dual-pol reflectivities and differential reflectivity
is also much smaller than any other moving conditions.

405



Furthermore, with this preliminary study, it was possible to identify other WT configurations, which are causing quite different polarimetric signatures with respect to the simple still WT condition, labelled with a) here below :

a) zero rotor speed and probably no change in either blade angles or nacelle orientation. Surely from 17:08 to 17:10 UTC (see Sec. 3.1); however, this configuration has probably been lasting until 17:13:31 UTC (see Sec. 3.2). Our hypothesis is that it has happened again in other two intervals: between 17:14:41 UTC ad 17:17:37 UTC (see Sec. 3.2) and during the last 40 s before 17:40 UTC (see Sec. 3.4), as it can be deduced from Figures 2 to 5.

b) Another peculiar (and probably rare) configuration is the one described in the central part of Sec. 3.2 as well as in Fig. 6 and that has occurred between 17:13:31 and 17:14:41 UTC. It could has been caused by a change in the blade angle, while the rotor speed was still 0. This is just an plausible hypothesis.. Whatever the reason could be, the changes in the differential backscattering coefficient, $\delta_{co}$, is impressive (see Fig. 6).

c) Then the most usual configuration comes, which is the one of energy production under sufficient wind conditions. We think that it has been lasting approximately 6 minutes (say from 17:22 to 17:28 UTC) during which most of the 22.5 rotations of the 17:20-17:30 interval has occurred.

d) Finally a configuration that is associated to large variability of the parametric signatures (from 17:17:37 UTC to approximately 17:22 UTC and, most of all, from approximately 17:28 UTC to 17:39:20 UTC, see Sec. 3.4).

Regarding a), we conclude that when the rotor speed is zero, the WT signatures are similar to those of a Bright Scatterer: we have observed, in fact, a very good stability and very small dispersion of the polarimetric variables; the situation is even better than what have been observed with a rotating antenna ($18°s^{-1}$) by Gabella (2018) using the metallic tower on Cimetta at 18 km range from the Monte Lema C-band radar. The even larger stability and small dispersion in the present campaign is due to the stare mode antenna of the X-band radar. A very small dispersion of the polarimetric variables is also observed during the first seven minutes of the following 10-min interval (from 17:10 UTC to 17:17:40 UTC, to be precise) and during the last 20 s of the 17:30-17:40 UTC interval. Our guess is that in both cases the rotor speed was equal to zero, which is exactly the status of the previous (17:00-17:10 UTC) and following (17:40-17:50 UTC) 10-min interval. Please note that we are not claiming that IF $\rho_{HV}$ is perfectly stable and always equal to 1, THEN the WT rotor speed must be zero. Rather, that IF the rotor speed is zero , THEN $\rho_{HV}$ is equal to 1, as long as no changes in orientation nor blade angles have occurred. From this viewpoint, it is worth focusing again on the 1000 radar polarimetric values acquired in 64 s from 17:13:31 UTC to 17:14:35 UTC (see Fig. 6): large changes in both horizontal and vertical polarization reflectivity (and consequently extreme variability of $Z_{dr}$ reaching an extreme maximum of +30 dB and a minimum of +4.5 dB); transition of $Z_{dr}$ from an unexpectedly large equilibrium value of +16 dB to a more "easier to understand" equilibrium around 0 dB; transition of the differential phase shift, $\Psi_{dp}$, from around ~0° to -466°, probably caused by an overall change of 466° of the differential backscattering phase shift. What can be the cause of such simultaneous large changes in $Z_{dr}$ and $\Psi_{dp}$? Blade angles? Nacelle orientation? Both? Or there was also a small movement of the rotor? It is hard, if not impossible, to find an answer to such questions with the present data. For future





campaigns, it is  obvious to recommend a much better (smaller) sampling time regarding the WT status and wind information:

ideally, from the current 600 s down to 1 s! Another obvious recommendation is related to the quantization of $\rho_{HV}$: either using

two bytes or a Log-transformation, like for instance the operational one used at MeteoSwiss (see Eq. 6 in Gabella, 2018).

There are two other worth mentioning facts: the first one is a sort of "intrinsic" inverse correlation between the dispersion and

the central value of the copolar correlation coefficient among many consecutive 64 ms echoes . When $\rho_{HV}$ is close to the

asymptotic value of 1, then the changes among successive echoes tend to be very small (see Fig. 3, from 17:30 to 17:40 UTC,

partial rotor rotation of 216˚ in ten minutes); as stated, when the rotor is not moving, blade angles and orientation not changing,

then $\rho_{HV}$ is consistently and constantly equal to 1 (see Fig. 3, from 17:08 to 17:13:31, as well as the detailed description in

Sec. 3.2). From 17:20 to 17:30 UTC (22.5 rotor revolutions), $\rho_{HV}$ varies widely, with a (8 s) minimum value often smaller than

0.1 and a median value lying between 0.7 and 0.8.

The second fact is an occasional, short lasting, quite surprising correlation between the differential phase shift (4th measurable,

see Sec. 2.3.4) and the differential reflectivity (2nd measurable, see Sec. 2.3.2) associated to a sort of cyclo-stationarity

(although during very short intervals) : this fact can be seen, for instance, during the 8.96 s (140 echoes) displayed in Fig. 7

during which approximately 5 periodic cycles of the two polarimetric measurable have took place. In Fig. 7, the vertical lines

are every 28 echoes (1.792 s); obviously, our intention is not to claim that the period is exactly 1.792 s, since 1.728 s (27

echoes) is certainly another reasonable estimate.

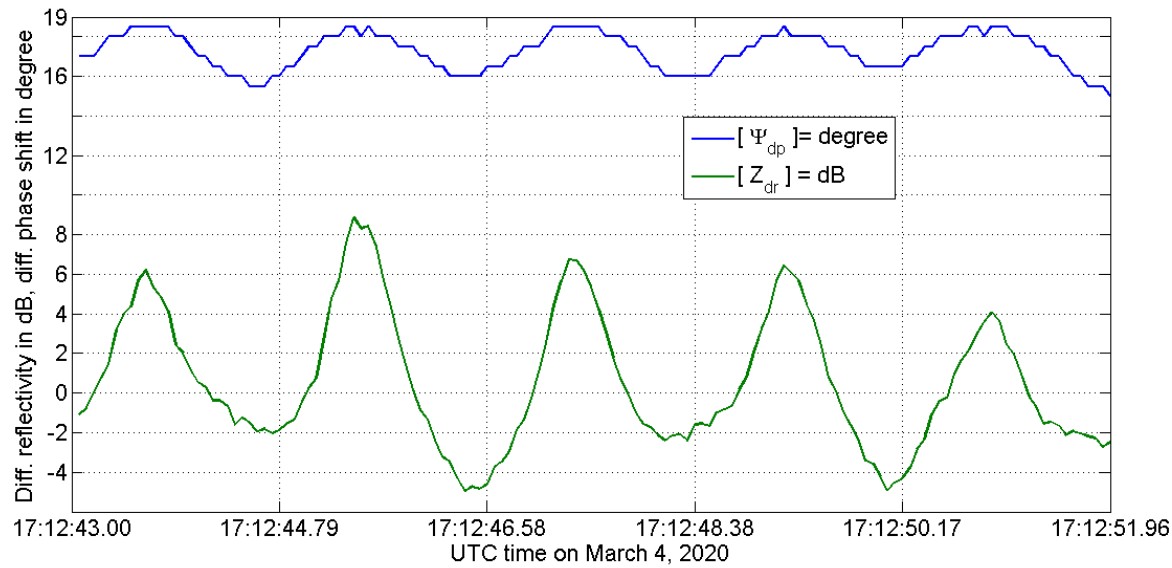


**Figure 7.** An example of quasi-cycle-stationarity of both the differential reflectivity in dB (green line) and the differential phase shift in degree (blue line) at the highest available temporal resolution, namely 64 ms. The abscissa spans an interval of exactly 8.96 s, which corresponds exactly to 140 echoes. Vertical lines are every 1.792 s, which is 28 consecutive echoes.



We think it is interesting to emphasize that there must be something WT-related with a period of ~1.7-1.8 s, which is reflected in both the (differential) phase and (differential, squared) amplitude of the polarimetric signals received by the radar.

**5 Summary, conclusions and outlook**

This technical note has extended the analysis and investigation by Lainer at al. (2021) in two directions:

1. To complement the statistics of horizontal polarization reflectivity, with those corresponding to other polarimetric
465       measurables: the copolar correlation coefficient, $\rho_{HV}$; the vertical polarization reflectivity; the differential reflectivity, $Z_{dr}$, which is defined as the Log-transformed ratio between horizontal and vertical polarization reflectivity; and the differential phase shift, $\Psi_{dp}$, between the phases of the copolar signals at horizontal and vertical polarization.

2. To investigate their variability at the best available temporal resolution (sampling time as short as 64 ms), despite the precious and valuable ancillary data related to the wind turbine status being available only every 600 s.

We have tackled the challenging sampling time (600 s vs 0.064 s) problem by starting with a 10-min interval that was characterized by zero rotor speed (still wind turbine). In such peculiar case we have observed that

- $\rho_{HV}$ is perfectly stable and always equal to 1 (DN = 255).

- 38.5 dBz $\leq$ ZV $\leq$ 41.5 dBz, i.e. only 7 Digital Numbers are used; the standard deviation is as small as 0.725 dBz.

- The temporal variability of $Z_{dr}$ is identical to ZV! (just with the opposite sign, obviously). How comes? Well, because,
475       to our great surprise, ZH is always equal to 56.5 dBz (see Sec. 3.1).

- 4° $\leq \Psi_{dp} \leq$ 40° during a 2-minute interval, with periodic oscillation of approximately ±3° in a bit less than 2 s; the standard deviation is as small as 2.9°.

The large difference in sampling time (64 ms vs 600 s) poses certainly a challenge to future analyses of the 3-week valuable campaign in March 2020. Nevertheless, we plan to extend this preliminary (32-minute) analysis (based on thirty thousand
polarimetric measurables) to another day (March 19, 2020) characterized by several 10-min intervals with zero rotor speed. The "prevailing in time", ~90-min long, stare mode acquisition of the 2020 campaign has been proved to be highly beneficial for a better characterization of the polarimetric signatures of the wind turbine, especially when it is still (or quasi-still). The results from previous studies (Lainer et al. 2021, Angulo et al. 2015) are, in fact, confirmed: the rotor speed is a key information in order to predict the values and the variability of backscattered power and phase of horizontal and vertical polarizations.
Another important parameter is the rotor blade angle (pitch), which is probably changed in a relatively short time, much shorter than the 600 s sampling time of the turbine data obtained so far. At the moment, more difficult to assess is the dependence on the nacelle orientation. Surely, we are just at the beginning of the fascinating task of deriving spectral and polarimetric signatures of wind turbines from the point of view of a weather radar not only in stare mode, but most of all with a rotating antenna.




*Data availability.* The data used in this study is available on request.

*Author contributions.* Conceptualization, data analysis and investigation, first draft preparation and writing: MG. Writing-review, discussions, interpretation and editing JG, ML, DW and MG. All authors have read and agreed to the submitted version
of the manuscript.

*Competing interests.* The authors declare that they have no conflict of interest.

*Acknowledgements.* As stated in the Introduction, this preliminary study regarding the polarimetric signature of a WT was
stimulated by comment Reviewer 1 (Interactive Comment, 2020), whom we would like to thank again. We would like to thank
Dr. Maurizio Sartori for having drawn Figure 1 and stimulating discussions, as well as Dr. Peter Speirs for helpful discussions.
Further, we would like to thank Hegauwind GmbH & Co. KG Verenafohren, which kindly provided the operational data of
the wind turbines and Reto Pauli, who has advised our team on all kinds of wind turbine aspects.

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
