# Peer review of "On the polarimetric backscatter by a still or quasi-still wind turbine"

_Atmospheric Measurement Techniques, 2022_

## Author Comment (AC1)

amt-2022-316 Authors answers to Reviewer 1 comments.

**General Comments**

As with the earlier paper from Lainer et al. (2021) it was a fun for me to read this manuscript. It is the first time I find my review to be cited in the next publication :-).I strongly recommend to publish its content. Nevertheless, some improvements are necessary and among them are important issues.

We warmly thank the Reviewer for their positive feedback and appreciation of our work. We answer below point-by-point to the concerns and suggestions.

The most important issue is the interpretation of the presented measurements. I do not see anything "peculiar". Let me describe my point of view a bit more detailed; A WT is a scatterer that is neither small compared to the radar wavelength nor is it small compared to the diameter of the main lobe of the radar beam. It is - in general - not of constant shape but changing its properties with (i) nacelle orientation, (ii) rotor angle, and (iii) blade angle. The shape of the rotor blades even changes with (iv) wind speed, as the blades are bended by the wind. The echo "seen" by the radar further depends on the (v) elevation under which the radar "looks" at the WT and the exact (vi) height and (vii) horizontal position and (viii) the diameter of the radar beam at WT position. Furthermore, the (ix) position of the WT within the recent range gate of the radar has to be considered. --- There are more but minor dependencies that impact the echo from a WT, as the distance between radar and WT which is implicitly included in (v) to (ix) but further indicates how well the radar beam can be approximated by a plane wave.

We thanks the Reviewer for this relevant and well-phrased input, which is kindly appreciated. Indeed, we find this clear and exhaustive scheme very helpful for a better understanding not only of the present results; of which we are confident, but also for a future, exhaustive and detailed analysis of longer temporal intervals during which the WT rotor speed was zero (e.g., on March 19).

Regarding March 19, following the input of the Reviewer we conducted a preliminary analysis of 93 minutes (nine 10-min intervals plus one lasting 3 minutes and 5 seconds) characterized by zero rotor speed, which confirms the main results of our short paper. Most of all, it confirms also your interesting above-listed interpretation scheme. The last page of the present document presents a short summary of this preliminary analysis. Because the results agree with our current findings we decided not to include them in the revised version extensively, in order to keep the message of the manuscript as easily conveyable as possible and to keep the paper short.

To give a more intuitive description I cite an engineer who once told me: Imagine the WT was coated with polished chromium and you light the WT with a spotlight. You see the reflections gliding over the surface of the WT, occurring and vanishing with the motion of the WT. At visible wavelengths the surface of a WT is mat but at radar wavelengths it appears to be glossy.

We thank very much the Reviewer for this intuitive and illustrative description. We are glad that Reviews are public in this journal, so that they are available to the readers.

For the antenna we call the dependency on azimuth and elevation its directivity pattern. We know, the larger the antenna the stronger the (possible) gradients of the directivity pattern. For the scatterer the corresponding term is "differential scattering cross section." Which, in the end, is nothing else but the directivity pattern of a scatterer. The differential scattering cross section of a WT is at least(!) dependent on the nine parameters mentioned above (i to ix). As the WT is much larger than the radar antenna, we have to expect very strong gradients of the differential scattering cross section to occur.

The presented study investigates variation due to the first four parameters, keeping all radar related parameters constant. The stability of the echoes during periods where the WT is standing still (condition "a" in the discussion) indicates that WT and radar are very reliable. The variations of the echoes of different "type a" periods simply show the

dependency on rotor angle and blade angle. As these two angles are random but constant the measured values are random but constant.

For a slow rotating rotor the experiment measured the differential scattering cross section at high resolution, mostly regarding rotor angle. We see all the extreme values. With increasing rotational speed (and constant temporal resolution) the angular resolution at which we see the crossection is reduced/coarsened. Thus the extreme values are smoothed out, everything looks smoother. This is immediately seen in the figures.

On the other hand: rotor speed is totally unimportant for an instantaneous (single) radar beam and its echo. The integration over several pulses (here 128) introduces changes in the echoes due to rotational speed.

*Again, very kind of you to share with us your explicative interpretations of the figures: we are glad that Reviews are public in this journal, so that they will be available to all the readers.*

There is nothing peculiar but the scattering cross section of a WT is complicated. So, please, shorten the title and remove the term "peculiar". (E.g.: "On the polarimetric backscatter of a still or quasi-still wind turbine.")

*We entirely agree with this comment (and with Reviewer 2): we have shorten the title following this suggestion.*

Dealing with the partially very precise time information is difficult and inconvenient. I propose to add two different indicating schemes:

1. Mark the four 10-min periods for which you have WT properties as I to IV in the figures. (Introducing e.g. black vertical markers at 17:10, 17:20, 17:30 and creating the four different "WT time steps".)
2. Mark those periods with comparable rotational speed and blade angles as indicated as a) through d) in the discussion by e.g. blue vertical markers and indicate the periods as a_1, a_2, a_3, b_1, and so on.

*Thank you for the suggestion: indeed, we have "labelled" the "distinctive" periods of interest, also in each Sub-sec title. P1, from 17:08 to 17:10, which corresponds to a still WT; from P2 to P4, the successive three 10-min intervals.*

Most of the precise time indicators in the text could be replaced by these indications of time periods. The markers can occur in the figures 2 to 5. Figure 6 and 7 should then be assigned to the corresponding periods.

*We thank the Reviewer for the suggestion, that we included in the revised version of the manuscript A thick line in Figure 2 introduces the sub-period P2.a, which is shown in Fig. 7. P2.a  is thoroughly described in Sec. 4 "Discussion (despite it lasts only 8.96 s!)*
*Two "start" and "stop" markers are associated with P2.b, which lasts 80 s, are shown in Fig. 6 and is thoroughly described in the same Sec. 3.2 (rotor partial rotation equal to 72 deg and blade pitch angle changed from 70 deg to 65 deg).*

The authors expect the differential reflectivity to be close to 0 dB (line 435: "easier to understand"). If we recall that photographers use a polarizing filter to reduce reflections on (glossy) surfaces we know that reflections at (glossy) surfaces may introduce polarization effects. Especially, multiple reflections (internally, only from the WT) will cause strong polarization of the backscattered signal. (Review also Line 387 f.)

*Indeed, we are deeply grateful for this explicative and clarifying comment: we agree that the exclamation mark at line 388 is certainly misleading. Furthermore, this comment helped the authors to shorten al the three "bullet-sentences".*

**Minor remarks**

The abstract shows already very detailed information which is not necessary. If the authors insist on having these details in the abstract, they should add the distance between radar and WT.

*We agree with the suggestion and we have rewritten the abstract following the suggestions of both Reviewers.*

Gabella et al. (2008) (line 90), Gabella and Perona (1998) (line 92), and the book by Fabry (line 191) do not show up in the references. I did not check more entries but obviously the references have to be controlled.

*We apologize about the omission. We have checked the references in the revised version. We thank the Reviewer for spotting this issue.*

In line 108 it needs to be 180 m x 180 m x 75 m. We have added twice the units (m) after 180.

Line 182: remove one "that" Removed, following the suggestion of the Reviewer.

Line 373: red curve in Fig. 2 (not inf) "f" has been deleted.

Line 412: Shouldn't it be "It could have been caused"? Thank you for correcting my mistake.

Line 430: The comma is falsely shifted to line 431.We corrected the typo and thank the Reviewer for spotting it.

Line 452: Remove "have". Deleted

Line 473f: Use Z_v as introduced in 2.3.1 and not ZV. (Same for ZH) The variables appear now as $Z_h$ and $Z_v$, thank you.

ADDENDUM

PRELIMINARY ANALISYS REGARDING 92 MINUTES OF PERFECTLY STILL CONDITIONS ON MARCH 19, 2020.

From 3:30 UTC to 5:10 UTC, none of the 3 most relevant parameters for the backscatter have changed. Both nacelle orientation and blade pitch angle have remained the same. Most of all, the 10-min average (and even max.) rotor speed was constantly equal to 0. Not surprisingly, all 86250 $\rho_{hv}$ values were equal to 1 (DN=255).

Table 1: Values of $\rho_{hv}$ during 102 minutes on March 19

| UTC time | 10-min average rotor speed in m/s | 10-min average | 10-min median | 10-min MAX. | Sequential # of intervals |
|---|---|---|---|---|---|
| 03:20-03:30 | 0.01 | 0.9854 | 1.0000 | 1.0000 | 1 |
| 03:30-03:40 | 0.00 | 1.0000 | 1.0000 | 1.0000 | 2 |
| 03:40-03:50 | 0.00 | 1.0000 | 1.0000 | 1.0000 | 3 |
| 03:50-04:00 | 0.00 | 1.0000 | 1.0000 | 1.0000 | 4 |
| 04:00-04:10 | 0.00 | 1.0000 | 1.0000 | 1.0000 | 5 |
| 04:10-04:20 | 0.00 | 1.0000 | 1.0000 | 1.0000 | 6 |
| 04:20-04:30 | 0.00 | 1.0000 | 1.0000 | 1.0000 | 7 |
| 04:30-04:40 | 0.00 | 1.0000 | 1.0000 | 1.0000 | 8 |
| 04:40-04:50 | 0.00 | 1.0000 | 1.0000 | 1.0000 | 9 |
| 04:50-05:00 | 0.00 | 1.0000 | 1.0000 | 1.0000 | 10 |
| 05:00-05:02 | 0.00 | 1.0000 | 1.0000 | 1.0000 | 11 |

From 3:20 UTC to 3:30 UTC, only a partial rotation of 36 degree has occurred, which has caused several "drops" of $\rho_{hv}$ below 1. During this 10-minute period, the range of Zh (Zv) goes from 25 (35) dBz to (72.5) 67.5 dBz, as it can be seen in Table 2.

Table 2: Minimum and maximum values of the radar reflectivity factors during five 10-min intervals.

| UTC time | 10-min average rotor speed in m/s | $Z_h$ 10-min minimum | $Z_h$ 10-min Maximum | $Z_v$ 10-min minimum | $Z_v$ 10-min Maximum |
|---|---|---|---|---|---|
| 03:20-03:30 | 0.01 | 25.0 dBz | 67.5 dBz | 35.0 dBz | 72.5 dBz |
| 03:30-03:40 | 0.00 | 54.5 dBz | 55.5 dBz | 50.0 dBz | 52.0 dBz |
| 03:40-03:50 | 0.00 | 54.0 dBz | 55.5 dBz | 50.5 dBz | 54.0 dBz |
| 03:50-04:00 | 0.00 | 53.5 dBz | 55.0 dBz | 53.0 dBz | 55.0 dBz |
| 04:00-04:10 | 0.00 | 54.5 dBz | 55.5 dBz | 53.0 dBz | 54.5 dBz |

Finally, the figures in the next page show the minimum, median, average and maximum values of the radar reflectivity factor every 8 s during 50 consecutive minutes (see table 2 above ) for horizontal (top picture) and vertical polarization (bottom picture). Being the original sampling time 64 ms, 125 "echoes" have been used to derive such four statistical indicators, two for the central location and two for the envelope. In turn, each echo has been derived by the Radar Signal Processor using 128 pulses (128 I and Q values) for each polarization state.

---

## Author Comment (AC2)

amt-2022-316 Authors answers to Reviewer 2 comments.

**General Comments**

The topic of this study is both interesting and timely as the weather radar community seeks to find mitigating actions to cope with the increasing number and size of wind turbines. One of them is to understand how the wind turbines are seen in the observations and can this information be used to identify and remove the wind turbine echoes as simultaneously keeping the precipitation echoes. The polarimetric variables has generally proven to be a useful metric for the classification of clutter and there have not earlier been many studies of the polarimetric signatures, one comes to mind Hall et al. 2017, where the dual-pol variables are used to classify the wind turbine echoes at C-band with a fuzzy logic - based methodology.

I find this study important, and it provides new insights that can be used to develop classification algorithms. However, in my opinion, this manuscript and the presented research are not yet scientifically mature enough to be published, and I would recommend major changes. The authors acknowledge that these are preliminary investigations in the title, and I would encourage them to analyze their dataset more thoroughly to provide more conclusive results, including the second period of still wind turbines (March 19, 2020), which would strengthen their findings.

Thank you for your stimulating and encouraging comments and the helpful information regarding the interesting study of the polarimetric signatures by Hall et al., of which we were not aware. Yes, there is still potential in our dataset, for not only polarimetric measurables, but also regarding Doppler and Spectrum width. Not limited to still conditions (March 19), but also for the other days. We hope we will be able to exploit it thoroughly in future years.

Regarding March 19, following your stimulating input, we have conducted a preliminary analysis of 93 minutes (nine 10-min intervals plus one lasting 3 minutes and 5 seconds) characterized by zero rotor speed, which confirms the main results of our short paper. In particular, from 03:30 UTC to 05:10 UTC, there were ten 10-minute periods characterized by 0 rotor speed. As you will see in the preliminary analysis presented at the end of this document, they confirm the main findings of the present short note:

- $\rho_{hv}$ always equal to 1.000 (to be precise, DN=255, namely rhoHV larger than ~0.996); this means a 2-way biunivocal (rs=0 <--> $\rho_{hv}$=1) correspondence (so far) during eleven 10-min periods (ten on March 19, one on March 4).
- Very small dispersion and temporal variability of Zh, Zv and Zdr.

Because the results agree with our current findings, we decided not to include them in the revised version extensively, in order to keep the message of the manuscript as easily conveyable as possible and to keep the paper short.

As noted by Anonymous Referee #1, the title is quite complex, and I agree that it could be shortened. Additionally, the use of the word "peculiar" to describe the polarimetric signatures is confusing, and I suggest using another adjective, such as "distinctive." Checking the Merriam-Webster dictionary the word peculiar is defined as characteristic of only one person, group, or thing: DISTINCTIVE or different from the usual or normal: ECCENTRIC, UNUSUAL. I assume the authors have meant the first interpretation, but I and I assume also the other Anonymous Referee #1 interpreted the second option and it was slightly confusing to read the manuscript. The language throughout the manuscript should be checked for any phrases or words that are more appropriate for spoken language than written language. While I am not a native speaker, I can provide some examples in the minor comments section.

Thank you so much. It has been so kind of you and Reviewer 1 to clarify this important aspect and to help us in finding a better title. Indeed, we meant "distinctive" rather than "eccentric"; however, we think that the best way to avoid any confusion is to follow the suggestion of both Reviewers: avoid using the adjective "peculiar".

Thank you also for the patient help with the English language, as none of us is a native speaker, too. We appreciate very much your help and are grateful for the valuable suggestions in the "Minor comments" section at the end.

In the detailed theoretical explanations or definitions, such as in section 2.3, the authors should pay attention to using precise definitions. For example, the authors should distinguish between radar reflectivity and reflectivity factor, which

have different dimensions. I suggest providing an exact definition of these terms in the manuscript and then stating that the authors will use reflectivity to refer to reflectivity factor throughout the manuscript, as is common practice in the field.

In the manuscript, we have always used the 1-word term "reflectivity" for z instead of the more complete form "radar reflectivity factor". Yes, as stated in the manuscript [$z$] = mm$^6$/m$^3$, while using a Logarithmic transformation [$Z$] = dBz (see line 163 of the original submitted manuscript). Similarly, I have seen that in many papers/books the 1-word term "reflectivity" is used for η, which is the total equivalent backscattering cross-sectional area per unit volume. However, in the paper, there is no need to use η. [η] = m$^2$ / m$^3$. In Sec. 2.3.1 we have adopted your helpful suggestion above. Up to that point, we have always used the complete form "radar reflectivity factor". After that, in a few cases, we have used reflectivity, as it is common practice in the field.

**Major Comments**

As Anonymous Referee #1 noted, the abstract is way more detailed with the specific numbers. I would suggest rewriting the abstract by firstly providing a brief description of the measurement setup and then main conclusions without referring to specific periods.

We have rewritten the abstract following the suggestions of both Reviewers.

The suggestions of Referee #1 were good to clarify the periods of interest, the authors should name them and indicate them in the figures. Referring to the chosen names in the text will make the manuscript easier to read.

Thank you, yes, we have "labelled" the "distinctive" periods of interest following the suggestions of Reviewer 1: P1, from 17:08 to 17:10, which corresponds to a still WT; P2 to P4, for the successive three 10-min intervals, up to 17:40. The four labels "P1" to "P4" appear now also in the titles of the four subsections.

Line 42: The authors state that research on polarimetric signatures of wind turbines is rare, which is true. However, it would be helpful to see a comparison with at least one other study, such as the one by Hall et al. 2017

Yes, it would helpful to link some results of the 2019 campaign (both PPI and RHI, see Lainer et al. 2021) with the interesting results of Hall et al. 2017; while it is not at all straightforward to find such link with the stare mode part of the 2020 campaign, especially for the still and quasi-still conditions, which are investigated in this study and linked with observations of Bright Scatterers.

In lines 80-100, a schematic picture showing how the radar is located in respect to the wind turbine with the distances and stated elevation angles would be beneficial.

Thank you, this is a helpful comment. We forgot to explicitly to mention in the text Fig. 1.c from Lainer et al. (2021), which shows what you are asking. Now it is referenced in the manuscript.

As stated in general comments, especially the section 2.3.1 (lines 156-171) should be rephrased with correct terminology. The authors should be careful when using reflectivity and reflectivity factor, and "Log-transform" should be changed to e.g. "in logarithmic units." The lines 166-171: The authors should clarify the explanation of the range of reflectivity values that can be measured with Meteo Swiss radars, as it is currently unclear and DN is not defined.

We have adopted your helpful suggestion above ("stating that the authors will use reflectivity to refer to the radar reflectivity factor").

The authors should explain why they performed an extrapolation of 8 minutes in lines 251-253.

Thank you for this sensible and legitimate question. While trying to answer, I have realized it is a proper example of your relevant remark above: "a phrase that is more appropriate for spoken language than written language".

Consequently, we decided to delete it. Well, we have tried to express something similar at line 135 where we wrote "Unimportant if during 8 minutes no radar data are available …", (see your helpful comment below); also there, we have decided to delete the colloquial and unnecessary sentence.

In lines 296-302, the authors should rephrase the section and add references to the correct figures.

Yes, thank you very much, we apologize for this mistake. Obviously, we meant small dispersion of $\Psi_{dp}$ ; unfortunately, we wrote $\rho_{HV}$. Now it is corrected.

**Minor Comments**

Lines 51 – 64: Section about the BS. I do see the need to explain BS in general, but I cannot really see how a wind turbine could be used for monitoring hardware due to its varying signatures, at least operationally, and now the section reads as to justify the campaign set up by using then wind turbines as BS. Maybe considering rephrasing this section.

The former 51-64 paragraph has been shortened. However, please note that the paragraph does not intend at all to justify the campaign in view of a better understanding of BS. It rather intends to show that even the 10-min intervals with 0 rotor speed, which are apparently useless to characterize the typical signatures of an energy production WT (hence, a "rotating and moving" WT), can still be useful to a better understanding of BS.

Line 59: Clarify the meaning of "hit" in the context of the sentence, "However, since it is hit during the operational weather scan program…."

The sentence has been rephrased and the citation has been updated (the new link is now with a peer-reviewed Journal).

Lines 65 – 78: Rephrase this section and provide a brief description of the manuscript structure rather than providing detailed results.

These paragraphs have been rephrased and made shorter following your suggestion.

Line 100: Consider changing the word "peculiar" when referring to the stare-mode strategy.

Yes, and following your helpful comment and interpretation at the previous page, we have used "distinctive". An alternative would be to delete "peculiar" and use no adjective at all.

Line 116: Rephrase the title: "Wind turbine data and metadata collection: a very peculiar 40 min interval under detailed investigation." Rephrased.

Lines 121 – 123: When listing parameters, use "e.g." instead of "…", and write temperature with a lowercase letter.

Implemented, thank you.

Line 135: Rephrase "Unimportant if during almost 8 minutes no stare mode radar data are available" as it is unclear.

The sentence has been deleted.

Line 139: Use a lowercase letter for "maximum." We now use lowercase letter for "maximum".

Line 145: Rephrase the sentence to make it complete. Rephrased.

Check that equations are styled consistently throughout the manuscript.

Eq. 1 has been changed accordingly.

Line 185: Remove "very" in "A very important…." deleted.

Line 191: Clarify if the numbering "e06.1" is referring to a chapter.

Yes, "e06.1", which is the first part of the electronic supplement number six accompanying the book by Fabry. Clarified in the text, too.

Lines 208 – 211: Rephrase the example of quantization of co-polar correlation coefficient as it seems redundant and not necessary. Remove the extra "use" in line 209.

This is indeed a good suggestion. We have now rephrased it and shortened it.

Line 232: Clarify the meaning of "a standard deviation of $360°/12^{0.5}$ would be expected."

The standard deviation of a uniform distribution varying from 0 to B is by definition $B/\sqrt{12}$.

Lines 235-238: Rephrase and remove the detailed results in this section.

This is shortened, rephrased and the details moved now to a more appropriate section(Sec. 3.2).

Line 242: Suggested to use "radar variables" instead of "backscattering properties."

Great suggestion, thank you. We have used "polarimetric radar measurables", consistently with the title of subsection 2.3, line 154.

Line 251: Remove "amazing" as it is not typically used in a scientific context. Replaced.
Line 277: Remove "very" in "at the original (very high) temporal resolution." Deleted.
Lines 307 and 336: Replace "remarkable" as it is not typically used in a scientific context. Replaced.
Lines 308: Replace "huge" as it is not typically used in a scientific context. Replaced.
Line 371: Use a lowercase letter for "maximum." We now use lowercase letter for "maximum".
Paragraph 421 – 433: Remove "very" statements. The two "very" have been deleted.

Line 433 avoid using IF and THEN in capital letters in this section as it is not suitable for a scientific context, in my opinion.

If and then are now lower case.

In the Summary section, rephrase lines 472 – 477 without the exclamation marks and questions such as "How comes?" and "Well, because, to our great surprise."

Another good example of your relevant remark above: "a phrase that is more appropriate for spoken language than written language". Consequently, we have rephrased line 472-477.

In Figures and their corresponding captions (Figures 2. - 5.), could you clarify why "MAX" is written in capital letters while the other terms such as median, mode, and minimum are written in lowercase letters. I suggest to state somewhere in the text that the mean is not shown in these figures, since the text is often referring to mean.

Following your recommendation above regarding line 371, we have now used lowercase letter for "maximum".

In the introductory lines 240-244, it is stated that we are using the "central and most probable locations of the original 125 echoes available every 8 s: the median and the mode." Please note, that now the mode is omitted in the Figures 2. -5, also to make them visually clearer.

REFERENCES
Hall, W. et al. (2017), Offshore wind turbine clutter characteristics and identification in operational C-band weather radar measurements. Q.J.R. Meteorol. Soc., 143: 720-730. https://doi.org/10.1002/qj.2959

ADDENDUM

PRELIMINARY ANALISYS REGARDING 92 MINUTES OF PERFECTLY STILL CONDITIONS ON MARCH 19, 2020.

From 3:30 UTC to 5:10 UTC, none of the 3 most relevant parameters for the backscatter have changed.
Both nacelle orientation and blade pitch angle have remained the same. Most of all, the 10-min average (and even max.) rotor speed was constantly equal to 0. Not surprisingly, all 86250 $\rho_{hv}$ values were equal to 1 (DN=255).

Table 1: Values of $\rho_{hv}$ during 102 minutes on March 19

| UTC time | 10-min average rotor speed in m/s | 10-min average | 10-min median | 10-min MAX. | Sequential # of intervals |
|---|---|---|---|---|---|
| 03:20-03:30 | 0.01 | 0.9854 | 1.0000 | 1.0000 | 1 |
| 03:30-03:40 | 0.00 | 1.0000 | 1.0000 | 1.0000 | 2 |
| 03:40-03:50 | 0.00 | 1.0000 | 1.0000 | 1.0000 | 3 |
| 03:50-04:00 | 0.00 | 1.0000 | 1.0000 | 1.0000 | 4 |
| 04:00-04:10 | 0.00 | 1.0000 | 1.0000 | 1.0000 | 5 |
| 04:10-04:20 | 0.00 | 1.0000 | 1.0000 | 1.0000 | 6 |
| 04:20-04:30 | 0.00 | 1.0000 | 1.0000 | 1.0000 | 7 |
| 04:30-04:40 | 0.00 | 1.0000 | 1.0000 | 1.0000 | 8 |
| 04:40-04:50 | 0.00 | 1.0000 | 1.0000 | 1.0000 | 9 |
| 04:50-05:00 | 0.00 | 1.0000 | 1.0000 | 1.0000 | 10 |
| 05:00-05:02 | 0.00 | 1.0000 | 1.0000 | 1.0000 | 11 |

From 3:20 UTC to 3:30 UTC, only a partial rotation of 36 degree has occurred, which has caused several "drops" of $\rho_{hv}$ below 1. During this 10-minute period, the range of Zh (Zv) goes from 25 (35) dBz to (72.5) 67.5 dBz, as it can be seen in Table 2.

Table 2: Minimum and maximum values of the radar reflectivity factors during five 10-min intervals.

| UTC time | 10-min average rotor speed in m/s | $Z_h$ 10-min minimum | $Z_h$ 10-min Maximum | $Z_v$ 10-min minimum | $Z_v$ 10-min Maximum |
|---|---|---|---|---|---|
| 03:20-03:30 | 0.01 | 25.0 dBz | 67.5 dBz | 35.0 dBz | 72.5 dBz |
| 03:30-03:40 | 0.00 | 54.5 dBz | 55.5 dBz | 50.0 dBz | 52.0 dBz |
| 03:40-03:50 | 0.00 | 54.0 dBz | 55.5 dBz | 50.5 dBz | 54.0 dBz |
| 03:50-04:00 | 0.00 | 53.5 dBz | 55.0 dBz | 53.0 dBz | 55.0 dBz |
| 04:00-04:10 | 0.00 | 54.5 dBz | 55.5 dBz | 53.0 dBz | 54.5 dBz |

Finally, the figures in the next page show the minimum, median, average and maximum values of the radar reflectivity factor every 8 s during 50 consecutive minutes (see table 2 above ) for horizontal (top picture) and vertical polarization (bottom picture). Being the original sampling time 64 ms, 125 "echoes" have been used to derive such four statistical indicators, two for the central location and two for the envelope. In turn, each echo has been derived by the Radar Signal Processor using 128 pulses (128 I and Q values) for each polarization state.

---

## Referee Report (RR1)

Review on: "On the polarimetric backscatter by a still or quasi-still wind turbine" by Marco Gabella, Martin Lainer, Daniel Wolfensberger, Jacopo Grazioli

It is still my opinion that this manuscript should be published. In terms of its  content, I do not have much more to criticize about the present manuscript. Only a few small things still remain that somewhat limit the readability. In addition, I still found some typos.

The more important issues are:

– There are three angles that define the orientation of a rotor blade. Without a sketch it is difficult to make clear which angles are meant. "Nacelle orientation" is quite easily understood. Nevertheless, you did not specify what is meant by a "relative" nacelle orientation. The second angle is normally the "blade pitch" or "blade pitch angle". You call it hte "blade angle", but this could also be the angle of the blades rotation (being e.g. 0° when pointing vertically upwards, 90° when pointing horizontally, 180° pointing vertically downwards, and 270° when pointing horizontally to the other side). Furthermore what is a blade pitch angle of 0° and one of 90°? I assume 90° is the feathered position of the blade. – Please, introduce a clear description.

– I'm still missing the interpretation that during (comparable) fast rotation of the WT's rotor the radar measurement are averaged over a larger rotation angle. This leads to a more stable mean value of Zh and Zv while the range from minimum to maximum individual measurement is hardly reduced.

On the other hand I'm missing a remark on the very high reproducibility of the measurement as long as the rotor is not moving. It proves how reliable your measurements are. There is (nearly) no unexplainable, external noise but variations are reliably representing changes in the measured signal from the WT.

Both points are important as no operational radar can see what you measured in this experiment. It was fundamentally to keep the radar beam orientation fixed. With a scanning antenna you always see changes due to both: the WT movement and the radar antenna movement.

– I still have my difficulties with all these time information. Please, name the time periods and/or the times an refer to these names. There is no use to call 17:10 to 17:20 period P2 and then, two lines later, you again write 17:10 to 17:20 instead of P2. In line 412 the "sufficient wind condition" lasts von 17:22 to 17:28, line 415 uses the same times, but in line 354 you refer to 17:23 to 17:28 – and I think you talk of the same period. There are more then 50 references to some point in time within the manuscript. Each demands the reader to find that time in the figures; sometimes in more than one. From my point of view, there are 7 main periods: from (i) 17:08 to 17:13 (still WT), (ii) P2b, (iii) 17:14:40 to 17:17 (still WT), (iv) 17:17 to 17:23 (slow movements), (v) 17:23 to 17:28 (fast movements), (vi) 17:23 to 17:39:20 (slow movements), and (vii) 17:39:20 to 17:40 (still WT). You may (and do) subdivide these periods in smaller details (especially Figs 6 and 7). But please, reduce the number of indicated times significantly.

Minor remarks:

line 82: Type in the position. It is 47.700° and 8.664°

line 110: Is it important that the electromagnetic field is not planar? It is

nearly because the opening angle of the antenna beam is only 1.3°. The more important point is, that the surface of the WT is not "planar".

line 146: Please point out, that the rotation of 72° is not continuous during the 10 min interval.

line 163: "... measure two values that are orthogonal"? The values are not orthogonal. The corresponding polarization planes are.

line 168: Did you introduce DN? If not, you should not make use of it.

lines 188f: You did not introduce HH and VV, so you should not use these terms.

lines 194ff: "... of the backscattered electromagnetic field within the radar sampling volume ... " is a wrong reference. The scattering took place in the radar sampling volume. The measurement took place in the radar.

lines 225ff: The enumeration is a repetition of what is given since line 214. It should be removed.

line 233: The standard deviation of an equally distributed angle between 0° and 360° is 360°/sqrt(12°). I recommend to rewrite as 60° sqrt(3).

line 245: "has already took place"? Shouldn't it be "has already taken place"?

line 271: Please mention, that the value of 56.5 dBz is a random result. The important point is the stationarity.

line 305: Fig .5 should be Fig. 5

line 321: "..$\Psi_{dp}$ was oscillation between 11° and +5°". Please, indicate if you meant −11° or +11°. When using the "+" for 5°, do it also for 11°.

line 341: "in the figure shown in sec 3.2". Are you talking of Fig. 6 of this manuscript?

line 343: maximum (green (not red)), minimum (blue (not cyan))

line 358: What is interesting in the fact, that Zh reaches the 3rd maximum?

lines 359f: "In correspondence... " This is no sentence. Additionally, I do not get, why you mix information on ZDR and Zh of the echoes before and after.

lines 371f: I do not find these ZDR values in the figures. What is wrong?

line 371: Both times have typos.

line 470: The given range of Zv values is again arbitrary. The information is the small variation. Additionally, you should emphasize here that a moving radar will never observe this persistence.

---

## Author Response (AR2)

amt-2022-316.R2 Authors answers to Reviewer 1 comments dated 10.6.2023 (2nd Revision)

It is still my opinion that this manuscript should be published. In terms of its content, I do not have much more to criticize about the present manuscript. Only a few small things still remain that somewhat limit the readability. In addition, I still found some typos.

The more important issues are:

There are three angles that define the orientation of a rotor blade. Without a sketch it is difficult to make clear which angles are meant. "Nacelle orientation" is quite easily understood. Nevertheless, you did not specify what is meant by a "relative" nacelle orientation. We think that it is enough to use the expression "orientation", since a schematic view of two wind turbines in relative nacelle positions of 0° and 270° with respect to the radar angle of attack is presented in Figure 3a in Lainer et al (2021). Consequently, we have deleted the adjective "relative" in the caption of fig. 1. Furthermore, we have added in this revised version of the paper (line 140) a reference specifying that we refer to the same configuration of Lainer et al. 2021.

The second angle is normally the "blade pitch" or "blade pitch angle". You call it the "blade angle", but this could also be the angle of the blades rotation (being e.g. 0° when pointing vertically upwards, 90° when pointing horizontally, 180° pointing vertically downwards, and 270° when pointing horizontally to the other side). Furthermore, what is a blade pitch angle of 0° and one of 90°? I assume 90° is the feathered position of the blade. - Please, introduce a clear description. Sorry, in this revised version we now use the expression "blade pitch angle" everywhere. Yes, according to what we have learned from Hegauwind GmbH, the blade pitch angle, $\Theta$, is the angle between the chord and the plane of rotation, so that it is close to 0 deg for average rotor speed, rs, larger than 6.5 rpm (most likely situation in Fig. 1, 12 hours) and around 70 deg for rotor rs≈0. In Fig. 1, a "rare", 3rd "intermediate" configuration state can be seen, see e.g., P3 (rs=2.25 rpm; $\Theta$=15 deg), P7 (rs=2.81 rpm; $\Theta$=28 deg) and P8 (rs=4.63 rpm; $\Theta$=27 deg). As stated, an explicit reference to the clear description of Lainer et al. 2021 has been added in this revised version of the paper at line zxc.

I am still missing the interpretation that during (comparable) fast rotation of the WT's rotor the radar measurement are averaged over a larger rotation angle. This leads to a more stable mean value of Zh and Zv while the range from minimum to maximum individual measurement is hardly reduced. We have added such interesting interpretation in Sec. 3.3, which is focused on period P3 (17:20-17:30); if we assumed that 30 rotations had occurred between say 17:24 and 17:28 (median rhoHV between 0.7 and 0.8), then one rotation would occur in exactly 8 s, which is exactly the interval of the running median (black dashed line), running minimum (blue) and running maximum (green) displayed in the plots.

On the other hand I'm missing a remark on the very high reproducibility of the measurement as long as the rotor is not moving. It proves how reliable your measurements are. There is (nearly) no unexplainable, external noise but variations are reliably representing changes in the measured signal from the WT. Both points are important as no operational radar can see what you measured in this experiment. It was fundamentally to keep the radar beam orientation fixed. With a scanning antenna you always see changes due to both: the WT movement and the radar antenna movement. Yes, indeed! We have modified/integrated the last paragraph of the manuscript accordingly.

In line 412 the "sufficient wind condition" lasts von 17:22 to 17:28, line 415 uses the same times, but in line 354 you refer to 17:23 to 17:28 - and I think you talk of the same period. Yes, sorry for such inconsistency: we have opted for changing (twice) 17:22 into 17:23, which seems more appropriate to us (according to Figs. 2-5).

I still have my difficulties with all these time information. Please, name the time periods and/or the times an refer to these names. There is no use to call 17:10 to 17:20 period P2 and then, two lines later, you again write 17:10 to 17:20 instead of P2. There are more then 50 references to some point in time within the manuscript. Each demands the reader to find

that time in the figures; sometimes in more than one. From my point of view, there are 7 main periods: from (i) 17:08 to 17:13 (still WT), (ii) P2b, (iii) 17:14:40 to 17:17 (still WT), (iv) 17:17 to 17:23 (slow movements), (v) 17:23 to 17:28 (fast movements), (vi) 17:23 to 17:39:20 (slow movements), and (vii) 17:39:40 to 17:40 (still WT). You may (and do) subdivide these periods in smaller details (especially Figs 6 and 7). But please, reduce the number of indicated times significantly. Yes, we have tried to reduce the number of indicated times; also Reviewer 2 has recommended to use the same time stamp format in the manuscript (see, e.g. lines 133-137, 145, 149, 151-154, 246, 255, 299, new lines 330, 337… , 351, 352, 363, 415). In particular, we have made Sec. 3.2 more concise. We are grateful to both Reviewers for their helpful and coherent suggestions.

**Minor remarks**

line 82: Type in the position. It is 47.700° and 8.664°. typo corrected.

line 110: Is it important that the electromagnetic field is not planar? It is nearly because the opening angle of the antenna beam is only 1.3°. The more important point is, that the surface of the WT is not "planar". We have rephrased the sentence: the focus is now on the not-planar, complex shape of the WT.

line 146: Please point out, that the rotation of 72° is not continuous during the 10 min interval. We have included this remark in the revised version of the manuscript.

line 163: "... measure two values that are orthogonal"? The values are not orthogonal. The corresponding polarization planes are. We have corrected the sentence, thank you.

line 168: Did you introduce DN? If not, you should not make use of it. Introduced.

lines 188f: You did not introduce HH and VV, so you should not use these terms. Removed in the revised version.

lines 194ff: "... of the backscattered electromagnetic field within the radar sampling volume ... " is a wrong reference. The scattering took place in the radar sampling volume. The measurement took place in the radar. We have rephrased the sentence. It is now shorter and simpler.

lines 225ff: The enumeration is a repetition of what is given since line 214. It should be removed. Removed.

line 233: The standard deviation of an equally distributed angle between 0° and 360° is 360°/sqrt(12). I recommend to rewrite as 60° sqrt(3). This has been rewritten. Now the square root is at the numerator rather than at the denominator.

line 245: "has already took place"? Shouldn't it be "has already taken place"? Mistake corrected, thank you.

line 305: Fig .5 should be Fig. 5; Typo corrected ("switching the blank")

line 321: "..\Psi_{dp} was oscillation between 11° and +5°". Please, indicate if you meant -11° or +11°. When using the "+" for 5°, do it also for 11°. Yes, sorry, from +11 to +5 deg.

line 341: "in the figure shown in sec 3.2". Are you talking of Fig. 6 of this manuscript? Yes, sorry, I am exactly referring to Fig. 6; I have rephrased line 341 accordingly.

line 343: maximum (green (not red)), minimum (blue (not cyan)) Absolutely right: sorry for not having updated the color correspondence. Note that, following the suggestion of Rev. 2, we have opted for moving such info directly into the figure captions.

line 271: Please mention, that the value of 56.5 dBz is a random result. The important point is the stationarity. Excellent suggestion, thank you: this important observation has been added to the text.

line 470: The given range of Zv values is again arbitrary. The information is the small variation. Additionally, you should emphasize here that a moving radar will never observe this persistence. Same as above (text at line 271).

line 358: What is interesting in the fact, that Zh reaches the 3rd maximum? As you pointed out in (the green emphasized) lines 271 and 470, these values are somehow arbitrary, too.

lines 359f: "In correspondence... " This is no sentence. Additionally, I do not get, why you mix information on ZDR and Zh of the echoes before and after. Indeed, as you properly pointed out in (the green emphasized) lines 271 and 470,

such values are arbitrary. Furthermore, you are right, there is no need to mix $Z_{dr}$ and $Z_h$ information. Consequently, we have deleted the $Z_{dr}$ related parts, thank you very much.

lines 371f: I do not find these ZDR values in the figures. What is wrong? There is nothing wrong. Since they do not represent neither the maximum, nor the minimum, nor the median value during 8 s, then nobody could find such "64 ms $Z_{dr}$ values" in Fig. 5, which displays the running MAX., median and minimum with a sampling time of 8 s. Three lines before line 371, we have now added in the text that such echoes have a temporal resolution of 64 ms (as in the line before 371, where we wrote "In both cases "In both cases the two 64 ms echoes are consecutive").
line 371: Both times have typos. Indeed, sorry about that. Now they are correct.

amt-2022-316.R2 Authors answers to Reviewer 2 comments dated 10.6.2023 (2nd Revision, "R2")

**General Comments**

The authors have adequately addressed most of the comments raised in the Reviewers' documents, resulting in an improved readability of the manuscript. I appreciate the efforts made to redraw the images, which are now clearer to the reader. The text has also become more lucid, and the manuscript's structure has been logically organized.

However, there remain a few minor issues that I would like to bring to the authors' attention, which I believe should be addressed prior to the publication of the final paper.

We thank the Reviewer for the positive feedback and for the additional suggestions and remarks that we have implemented in this further revised version of the manuscript.

**Minor Issues**

The new abstract version is improved, just a few comments:

-In line 16, it is stated that "the copolar correlation coefficient between the two orthogonal polarization states was persistently equal to 1." However, in lines 23-24, it is reiterated that "the copolar correlation coefficient between the two orthogonal polarization states, ρHV backscattered by a still Bright Scatterer should be equal to 1." To avoid repetition, it is unnecessary to include the phrase "between the two orthogonal polarization states" in the second instance. And I suggest to include the symbol ρHV the first time the variable is mentioned, which is in line 16. However, the last sentence in line 24-25, which states, "It is confirmed that the copolar correlation coefficient between the two orthogonal polarization states, ρHV backscattered by a still Bright Scatterer should be equal to 1 if observed by a non-rotating radar antenna," feels somewhat awkward. Actually, it might be more effective to consolidate the information and state it in line 16 as follows: "The copolar correlation coefficient, ρHV, between the two orthogonal polarization states was persistently equal to 1, similar to the signature of a Bright Scatterer (BS) observed by a non-rotating radar antenna."

This is indeed a good suggestion, which we have gladly implemented.

-Line 34: yes, we now use "strong", which is better than "heavy".

-Line 47: the link has been moved to the References, thank you.

Last paragraph in Introduction, starting from 68. I would move all the description or statements of results from this section either to discussion or abstract, but not in this section. I suggest rephrasing. Sorry, if possible, we would prefer to keep such introductory part that helps the readers orienting themselves depending on their specific interests; for instance, a weather radar expert interested in the signatures of the WT, would skip Sec. 2 and go directly to Sec. 3.

The manuscript is quite burdensome to read due to the rigorous indications to the time stamps. To help the reader I would suggest always using the same time stamp format, i.e. the chosen e.g. 17:20 UTC. For example, in line 133: "followed by a quiet period that was approaching in the last twenty minutes preceding 17 UTC", why not 16:40 – 17:00 UTC. Another example in lines 422-423. Please rephrase the whole document accordingly. We have rephrased the document accordingly, thank you for the suggestion.

Equation 2: I believe the division sign is accidentally marked as subscript? You are right, sorry. Now it is correct.

I have a general comment regarding the Figure descriptions in the manuscript. It would enhance readability if the information about line colors and dots in the Figures are included directly in the Figure captions, for example, in lines 148 and 243. Lines and dots color is now described both in the text and in the caption.

-Line 340. Missing space before "Similarly...". Space added.

-In lines 343 – 342 are referred to the colors of the old Figures. I suggest removing the color descriptions from the text and keeping them only in the captions as suggested above. Color description removed.

-In lines 351-352 are stated the mode and mean. Should these be removed or if not, maybe to add in parenthesis (not shown here). You are right, thank you very much. We have opted for removing such detail regarding mode and mean.

-In lines 406 and 468: The word "peculiar" is still used. We apologize for having forgotten to substitute peculiar with distinctive, as per your kind and helpful suggestion during Revision1. Now, it is done.
In line 409, "quite impressive", I suggest e.g. the word significant instead. Done, thank you.

-Please ensure that each abbreviation is clearly defined upon its first mention, for instance, DN, RCS, rs, HH, and VV. Actually, I don't think HH and VV are needed and as they are not used elsewhere, they could be removed. DN and RCS have been introduced. Rs, HH and VV have been removed.

-The paragraph starting 244 is already describing the first P1 period results. I suggest moving it to 3.1. If possible, we would rather keep it in such introductory part of Section 3, followed by detailed discussions in the sub-sections 3.1 to 3.4.

-The start and end of the P2a and P2b are plotted only to Figure 2. I would suggest adding them also to the Figures of the other variables. Now, we indicate the P2a and P2b periods in all four Figures (2-5).

Figure 2: "that contains the pole of the wind turbine" is confusing phrase. Does it mean that it doesn't include the blades? Thank you for the observation. Since such geometrical aspect is thoroughly described in lines 106-108, we have simply shortened the text by deleting "pole of the".

Spoken language is in lines 334-335 and the same questions are used again in lines 434. I suggest rephrasing. The paragraph (now lines 335-337) has been rephrased and shortened.

Figure 7. Are the lines e.g. maximum or mean, please specify. Fig. 7, just like Fig. 6, shows radar measurables at the original 64 ms sampling time (derived from 128 I and Q values acquired with a 2 kHz PRF).

---

## Author Response (AR3)

amt-2022-316.R2 Authors answers to Reviewer 1, comments dated 6.8.2023 (3$^{rd}$ Revision)

I had a lot of fun reading the improved version of the manuscript. I'm glad that a lot of time information was reduced by introducing named periods (P1 through P4). There are only few remarks and some technical issues left:

1) I'm missing a statement that the high stability of the measurements at the still turbine proves the high quality and low noise of the total system, including radar (transmitter, antenna, receiver) and wind turbine. Variations during WT motion are representing physical changes and are not caused by noise. One can rely on these measurements!
We have added it in the conclusion at line 467.

2) (Probably not for putting it into the manuscript): It looks like Z_h is more stable than Z_v for Z_v in P1. This is true in logarithmic scales. But when transforming back to linear reflectivity values, the range between 56,75 dBZ and 56,25 dBZ is a delta of 52000 mm$^6$/m$^3$ whereas the range from 38,5 dBZ to 41,5 dBZ is only 7000 mm$^6$/m$^3$. So, there is no prove that Z_v is varying stronger than Z_h.
Interesting viewpoint, thank you: I see you evaluating the variability in an "additive" way, I mean using a linear scale. Personally, I would evaluate it in a "multiplicative" way, which is using Log-transformed values. As you suggested, it is probably not worthwhile to discuss such different viewpoints inside the manuscript.

**Technical points**

I 73: Those constructive and destructive interference maxima occur also under conditions of a rotating WT rotor. But they are less visible as an average over a wider rotation angle range is determined.
We have rephrase the paragraph accordingly.

I 79: Neely et al is missing in the literature. Please, check if all cited literature is given in the bibliography and if all literature in the bibliography is cited in the text.
We went through the whole references and bibliography to check this issue. In the end we opted for leaving out the citation of Neely et al.

I 129: I do not see 7 to 11 rpm (only 7 to 8 rpm in Fig. 1).
Yes, the first half of the day (0-12 UTC) is not shown in Fig. 1, which covers only the second half of the UTC day (12-24 UTC); we have modified the text.

I 132: Please, indicate that 17:00 through 17:10 is P1 and can be found in Fig. 2. Otherwise one is astonished to see P2 quoted in I 144.
It is now indicated.

Figure 1: blue (instead of blu), typo corrected.

I 218: I know, I am petty, but: PsiDP, especially PhiDP are not a property "at any given range" but they are (normally) a property of the atmosphere between the radar and range r (plus backscattering phase plus Psi0). Could we say "PsiDP _of_ any given range"? I think "_at_" is wrong.
Well, none of us is native speaker. Since the dependence of the range, $r$, is not explicitly shown in eq. 3 [we simply write $\Psi$, not $\Psi(r)$], I have opted for removing "at any given range".

l 289: Fig. 5 (not 4) for ZDR, typo corrected.

l 300: Fig. 6 (capital F), typo corrected.

l 304: This is not the first relative minimum (but the third). Please, find a better formulation. We have deleted "first".

l 312: ZDR is only "expected" to be 0 dB for a rotating WT. As long as the WT is standing still both reflectivities (Z_h and Z_v) show random values and thus ZDR is "twice as random". 0 dB is the expected average and is achieved after averaging over sufficient wide rotation angles of the WT.

Yes, indeed: 0 dB could be expected only by averaging several rotations of the WT. Actually, as you pointed out, Zdr is "twice as random": hence, 0 dB is not at all expected! We have deleted "expected".

l 316: I probably would add: These data show that already comparable small changes of the state of the WT can introduce significant impact on the radar observables.

We have further elaborated your valuable suggestion; our proposal is: "these data show that even at zero rotor speed, other changes of the state of WT can have a large impact on the radar measurables".

Figure 6: Could you add behind "(from 17:13:31 to 17:14:51 UTC" , P2b)?

Yes, please! Thank you. Similarly, we have specified in the caption of fig. 7 that the 140 echoes shown (8.96 s) belong to the selected interval "P2a".

l 346: It is a few tens of seconds before 17:40 UTC (i.e. starting at 17:39:41), yes, and maybe we do not need to list the exact time when, for instance, rhoHV went to 1; hence, we have opted for "around 17:39:40.

l 354: larger fluctuations of the median, maximum, and minimum reflectivity, "median" added.

l 356: Shouldn't it be Around 17:39:40 UTC?

Exactly so, thank you; and it has helped me to find another "bug" (see line 365 below).

l 357 and 359: These bounding values for Z_h and Z_v are random values. There is no scientific reason behind them being larger or smaller than the median average achieved during WT operation mode.

Yes, you are right: we have added "randomly" (and put the less relevant part between brackets).

Figure2, line 292 and more: The stable period is happening from 17:39:40 on. This is the last 20 seconds before 17:40. This period is cited several times and its duration is as often called 20 s as 40 s. In l 292 it is even given as the period before 17:30. Please, give a unified time and duration for this event.

You are right: I apologize for this error, which has occurred more than once; it is me, MG, who should be blamed. It looks like if I had in mind 17:32 instead of 17:40, as well as "20 s past" instead of "20 s to" (see also you 4 additional helpful comments below)… I am really sorry about this; thank you for your patience.

l 365: not 40 s but 20 s (see the given times behind).

Corrected. We apologize, also the time was wrong: the last 20 s of our study period ends at 17:40, not at 17:32!! (see also, your precious comment at line 356) Thanks to you, we could find this wrong time reference.

l 383: not 40 s but 20 s (see l365), corrected.

l 391: 17:23 to 17:38 is only 5 minutes, not 6; typo corrected, thank you.

l 399: shouldn't it be: "due to the antenna stare mode of the x-band radar", yes, thank you for pointing it out.

As you already followed my wish to further investigate your data I want to formulate some more ideas:

1) For a reduced data set of only "energy production mode" (i.e. pitch angle 2°) the mean radar observables could be determined as a function of nacelle orientation. This is the only varied parameter as long as you measure in stare mode, as pitch angle (blade angle) is fixed and variations due to rotation of the blades are averaged out.

2) The opposite data set of non-rotating rotor can be evaluated for extreme values achieved and on the gradients that occur due to variations in nacelle and pitch angles. Under still standing rotor and radar antenna you sample the "directivity pattern" of the WT (i.e. the differential scattering cross section) with a very high resolution (although you cannot vary all possible parameters, i.e. the radar is at a fixed position). We could learn about the strongest values of the differential scattering cross section and the "beamwidths" of these strongest values. My assumption is, this cross section looks more like an irregular "moravian star" than like a smooth "potato".

Thank you very much for these two interesting ideas for future investigations.